# Targeting the interaction between RNA-binding protein HuR and FOXQ1 suppresses breast cancer invasion and metastasis

Xiaoqing Wu [1,2], Gulhumay Gardashova [1], Lan Lan [1], Shuang Han[1], Cuncong Zhong [3], Rebecca T. Marquez[1], Lanjing Wei[4], Spencer Wood[5], Sudeshna Roy [5,13], Ragul Gowthaman [6], John Karanicolas [7], Fei P. Gao [8], Dan A. Dixon [1,2], Danny R. Welch [2,9], Ling Li[10], Min Ji [11], Jeffrey Aubé [5] & Liang Xu [1,2,12✉]

Patients diagnosed with metastatic breast cancer have a dismal 5-year survival rate of only 24%. The RNA-binding protein Hu antigen R (HuR) is upregulated in breast cancer, and elevated cytoplasmic HuR correlates with high-grade tumors and poor clinical outcome of breast cancer. HuR promotes tumorigenesis by regulating numerous proto-oncogenes, growth factors, and cytokines that support major tumor hallmarks including invasion and metastasis. Here, we report a HuR inhibitor KH-3, which potently suppresses breast cancer cell growth and invasion. Furthermore, KH-3 inhibits breast cancer experimental lung metastasis, improves mouse survival, and reduces orthotopic tumor growth. Mechanistically, we identify FOXQ1 as a direct target of HuR. KH-3 disrupts HuR–FOXQ1 mRNA interaction, leading to inhibition of breast cancer invasion. Our study suggests that inhibiting HuR is a promising therapeutic strategy for lethal metastatic breast cancer.

[1] Department of Molecular Biosciences, The University of Kansas, Lawrence, KS, USA. [2] The University of Kansas Cancer Center, The University of Kansas Medical Center, Kansas City, KS, USA. [3] Department of Electrical Engineering and Computer Science, The University of Kansas, Lawrence, KS, USA. [4] Bioengineering Program, The University of Kansas, Lawrence, KS, USA. [5] Department of Chemical Biology and Medicinal Chemistry, Eshelman School of Pharmacy, The University of North Carolina, Chapel Hill, NC, USA. [6] Center for Computational Biology, The University of Kansas, Lawrence, KS, USA. [7] Program in Molecular Therapeutics, Fox Chase Cancer Center, Philadelphia, PA, USA. [8] Protein Production Group, NIH COBRE in Protein Structure and Function, The University of Kansas, Lawrence, KS, USA. [9] Department of Cancer Biology, The University of Kansas Medical Center, Kansas City, KS, USA. [10] National Translational Science Center for Molecular Medicine, Department of Cell Biology, School of Basic Medicine, The Air Force Medical University, Xi'an, Shanxi, China. [11] School of Biological Science and Medical Engineering, Southeast University, Nanjing, Jiangsu, China. [12] Department of Radiation Oncology, The University of Kansas Medical Center, Kansas City, KS, USA. [13]Present address: Department of Biomolecular Sciences, School of Pharmacy, University of Mississippi, University, MS, USA. ✉email: xul@ku.edu

When diagnosed before metastasis, breast cancer is nearly always curable; however, if metastasis has already developed, the 5-year survival rate drops to 24%[1]. Therefore, the most important improvements in survival will result from either prevention of metastasis or better control of already disseminated disease. Identification of new therapeutic targets and development of potent inhibitors are urgently needed to combat lethal metastatic breast cancer.

The RNA-binding protein (RBP) Hu antigen R (HuR), also known as embryonic lethal abnormal vision-like protein 1 (ELAVL1), is a ubiquitously expressed post-transcriptional regulator[2]. Post-transcriptional gene regulation is essential for normal development, but when dysregulated, has many implications in disease conditions, such as cancer. HuR is broadly upregulated in virtually all malignancies tested[3]. Although HuR predominately localizes in the nucleus, its function of stabilizing and/or modulating the translation of target mRNA is linked to the translocation to the cytoplasm[4]. Cytoplasmic HuR accumulation is associated with high-grade malignancies with poor overall survival and disease-free survival, and may be an independent prognostic factor of poor clinical outcome in breast cancer[5–7]. HuR preferentially binds to mRNA bearing adenine- and uridine-rich elements (ARE), or uridine-rich sequences, typically located in the 3′-untranslated region (UTR)[8,9]. ARE is a specific *cis* element present in mRNA, which confers to rapid mRNA decay[10]. It is generally accepted that cytoplasmic binding of HuR to these ARE-containing mRNA leads to mRNA stabilization and increased translation by competing with decay factors in ARE[11,12]. Over the past two decades, numerous mRNA has been identified as HuR direct targets. These transcripts, which encode proto-oncogenes, growth factors and various cytokines, implicate in cell proliferation, survival, angiogenesis, immune recognition, invasion and metastasis[13]. Therefore, HuR is an emerging target for breast cancer therapy, especially for metastatic breast cancer.

HuR is reported to interact with the mRNA 3′-UTR of transcription factor Snail[14], metallopeptidase MMP-9[15] and serine proteinase uPAR[16]. Snail is responsible for the induction of epithelial-to-mesenchymal transition (EMT), while MMP-9 and uPAR are involved in extracellular matrix (ECM) degradation. Therefore, HuR is thought to promote invasion and metastasis by increasing expression of the proteins that induce the transition to a mesenchymal phenotype and degrade ECM. However, the specific molecular mechanisms underlying HuR effects on invasion and metastasis of breast cancer are not well understood.

We[17,18] and others[19–22] have sought to identify small molecule inhibitors that interfere with HuR–mRNA complex. These small molecules show moderate to high binding affinity to HuR in different biochemical assays and have been validated as HuR inhibitors[23]. However, only a few of them are potently cytotoxic to cancer cells and therapeutic efficacy of HuR inhibitors was only examined in bladder cancer xenograft model[24] and colorectal cancer xenograft models[25–27].

Here, we report the identification of a HuR small molecule inhibitor, KH-3. KH-3 potently inhibits breast cancer cell growth in vitro and in vivo. KH-3 inhibits breast cancer cell invasion in vitro as well as delays initiation of lung colonies and improves mouse survival in an experimental metastasis model in vivo. We also demonstrate that FOXQ1 is one of the downstream targets that contribute to HuR's role in breast cancer invasion. KH-3 suppresses breast cancer cell invasion by disrupting HuR–FOXQ1 mRNA interaction. Our data provide a proof of principle that HuR inhibition by KH-3 may be developed as a promising molecular therapy for inhibiting progression and metastasis of breast cancer with HuR overexpression.

**Table 1 Correlation between cytoplasmic HuR expression and the clinicopathologic factors (n = 135).**

| Parameter | | Cytoplasmic HuR, N (%) | | |
| --- | --- | --- | --- | --- |
| | | Low | High | P* |
| Age | <60 (y) | 53 (58.9) | 37 (41.1) | 0.189 |
| | ≥60 (y) | 32 (71.1) | 13 (28.9) | |
| Tumor grade | II | 67 (72.0) | 26 (28.0) | **0.002** |
| | II-III & III | 18 (42.9) | 24 (57.1) | |
| T stage | T1 | 33 (66.0) | 22 (34.0) | 0.590 |
| | T2 | 52 (65.0) | 28 (35.0) | |
| N stage | 0 | 46 (63.9) | 26 (36.1) | 0.860 |
| | ≥1 | 39 (61.9) | 24 (38.1) | |
| AJCC stage | 1 + 2 | 58 (64.4) | 32 (35.6) | 0.706 |
| | 3 | 27 (60.0) | 18 (40.0) | |
| Positive lymph node (n = 126) | <1 | 39 (59.1) | 27 (40.9) | 0.715 |
| | ≥1 | 38 (63.3) | 22 (37.7) | |
| Overall survival (n = 134) | Live | 72 (68.6) | 33 (31.4) | **0.010** |
| | Dead | 12 (41.4) | 17 (58.6) | |
| Relapse (n = 134) | No | 64 (66.0) | 33 (34.0) | 0.233 |
| | Yes | 20 (54.1) | 17 (45.9) | |
| Metastasis (n = 134) | No | 80 (65.0) | 43 (35.0) | 0.100 |
| | Yes | 4 (36.4) | 7 (63.6) | |
| Distant disease-free survival (n = 134) | Live | 70 (69.3) | 31 (30.7) | **0.007** |
| | Dead | 14 (42.4) | 19 (57.6) | |

Fisher's exact test.

## Results

**High cytoplasmic HuR correlates with poor clinical outcome.** To explore functional roles of HuR in breast cancer progression, we first initiated a retrospective study of HuR expression by immunohistochemistry staining of 140 breast cancer patient samples. Patients' clinicopathologic variables are summarized in Supplementary Table 1. As regulation of RNA stability and translation is mainly related to cytoplasmic localization of HuR, we focused on the cytoplasmic HuR expression. Cytoplasmic HuR was negative or low in 63.0% (85/135) and high in 37.0% (50/135) of 135 technically well-stained specimens. Representative immunostaining results are shown in Supplementary Fig. 1a. We then examined the association of cytoplasmic HuR expression with other clinicopathologic variables. As shown in Table 1, high cytoplasmic HuR was significantly correlated with high tumor grade, low overall survival rate and distant disease-free survival rate. Furthermore, 63.6% of patients with metastasis had high cytoplasmic HuR while 35.0% of patients without metastasis had high cytoplasmic HuR, though the difference did not reach statistical significance because of small number of patients with metastasis. These data suggest that patients with high levels of cytoplasmic HuR have higher risk to develop metastasis. Cytoplasmic HuR expression had no significant correlation with age, TN stage, AJCC stage, positive lymph node numbers and relapse.

Similar to univariate analysis in Table 1, when plotting the percentage of cytoplasmic HuR positively stained cells based on tumor grade, patients with grade II-III or III breast cancer had a significantly higher percentage of positively stained cells (Fig. 1a) compared to patients with grade II breast cancer. Patients with high cytoplasmic HuR expression had markedly lower overall survival rate (Supplementary Fig. 1b) and distant disease-free survival rate (Fig. 1b) compared to those with low cytoplasmic HuR expression. Taken together, these data suggest that elevated

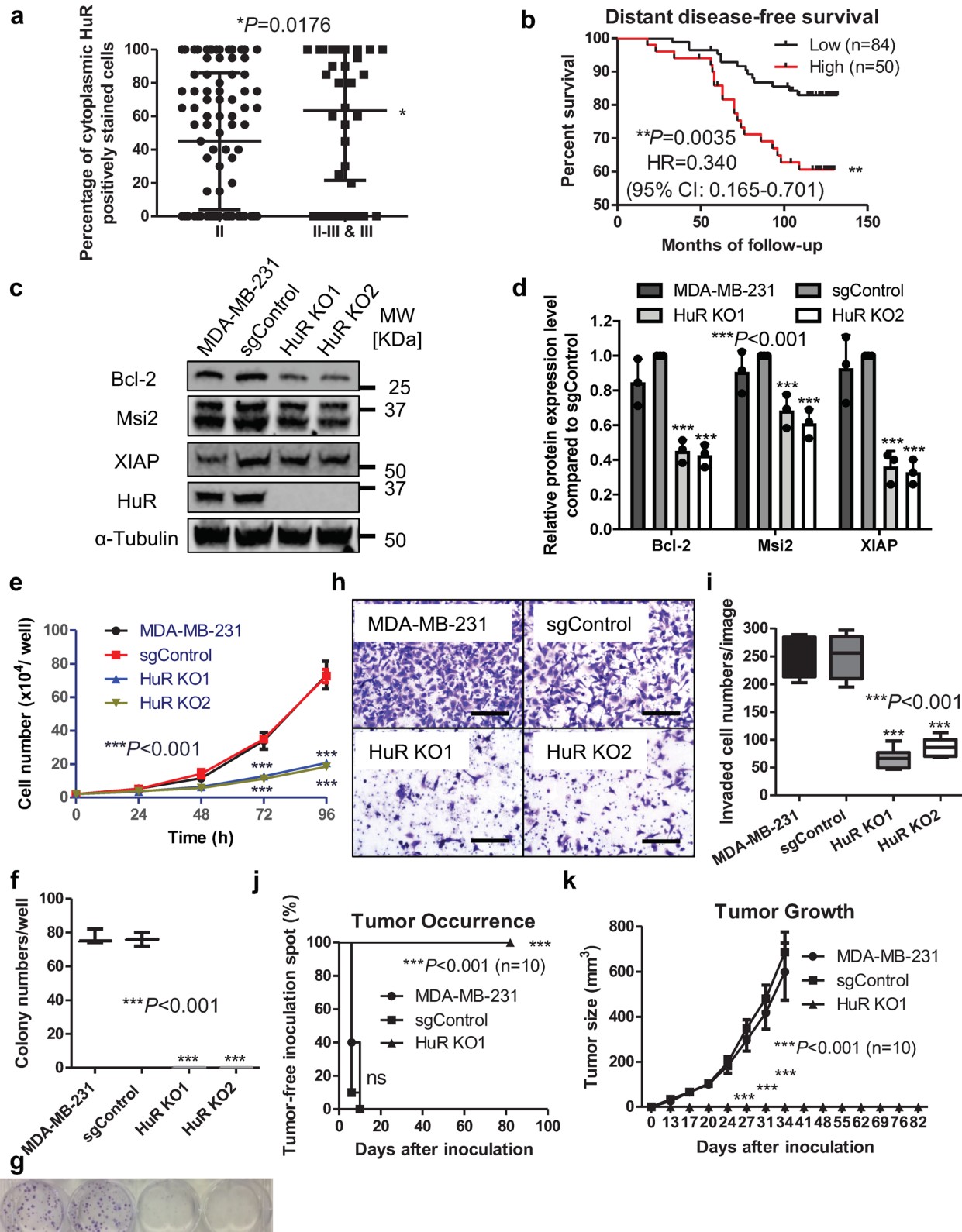

cytoplasmic HuR could serve as a diagnostic biomarker of higher tumor grade and prognostic marker of poor clinical outcome in breast cancer, consistent with previous reports[6].

**HuR inhibition decreases the proliferation and invasion.** We then manipulated the HuR expression level in a highly metastatic human triple-negative breast cancer (TNBC) cell line, MDA-MB-

231, using shRNA and CRISPR/CAS9 approaches to determine whether HuR inhibition would affect cell proliferation and invasion. Constant or inducible shRNA system reduced HuR protein level ~50% (Supplementary Fig. 2a, b), which resulted in significantly delayed tumor initiation and smaller tumor volume in vivo and fewer invaded cells in vitro (Supplementary Fig. 2c–f). However, cell growth rate and colony formation ability were not

**Fig. 1 HuR is a therapeutic target for breast cancer. a** Cytoplasmic HuR expression in breast cancer tissues with low or high grade. Patients with high-grade tumors (II-III or III, $n = 42$) have higher positively stained cytoplasmic HuR than those with low-grade tumors (II, $n = 93$) (*$P = 0.0176$, $t$-test). **b** Kaplan–Meier analysis of the distant disease-free survival of 134 patients comparing high and low cytoplasmic HuR. Patients with high cytoplasmic HuR have lower distant disease-free survival rate compared to those with low cytoplasmic HuR (**$P = 0.0035$, log-rank test). **c, d** Protein expression levels of HuR and downstream targets in MDA-MB-231 cells, cells with control sgRNA (sgControl) and two HuR KO clones. **c** Representative WB results from one experiment. **d** Quantified relative expression of HuR downstream target Bcl-2, Msi2, and XIAP. Values are mean ± SD from $n = 3$ independent experiments (***$P < 0.001$, two-way ANOVA). **e** Growth curves of MDA-MB-231 cells, sgControl and two HuR KO clones. Values are mean ± SD from $n = 3$ independent experiments (***$P < 0.001$, two-way ANOVA). **f, g** Colony formation of MDA-MB-231 cells, sgControl and two HuR KO clones. **f** colony numbers per well (***$P < 0.001$, one-way ANOVA, $n = 3$). **g** Representative images of colonies. **h, i** Invasion assay in parental MDA-MB-231 cells, sgControl and two HuR KO clones. **h** Representative images of stained invaded cells, scale bars: 200 μm. **i** The number of invaded cells per image (***$P < 0.001$, one-way ANOVA, $n = 6$). **j, k** Tumor initiation (**j**) and tumor growth (**k**) of MDA-MB-231 cells, sgControl and HuR KO1 clone in athymic nude mice. HuR KO1 is unable to engraft tumor in vivo ($n = 10$, ***$P < 0.001$, log-rank test for tumor initiation and two-way ANOVA for tumor growth).

obviously affected by knocking down HuR level (Supplementary Fig. 2g, h). In contrast, knocking out (KO) HuR expression using CRISPR greatly reduced cell proliferation and completely abolished colony formation. Two clones, HuR KO1 and HuR KO2, with depleted HuR protein level (Fig. 1c), also showed decreased protein expression levels of three known HuR targets Bcl-2[28], XIAP[29], and Musashi 2 (Msi2)[30] (Fig. 1c, d and Supplementary Fig. 3). HuR depletion was also confirmed by immuno-fluorescence staining using the same HuR antibody used for immunohistochemistry staining (Supplementary Fig. 4). The two HuR KO clones grew much slower than the parental cells and cells with control sgRNA (sgControl, Fig. 1e). In the colony formation assay, HuR KO clones were unable to form any colonies with 200 cells/well seeded, while parental cells and sgControl formed ~80 colonies per well (Fig. 1f, g). HuR KO clones also exhibited decreased invasion (Fig. 1h, i). Since HuR KO1 showed slightly less invasive capacity, we inoculated this clone as well as the parental cells and sgControl to the mammary fat pad (MFP) of female athymic mice to examine the tumor formation. In contrast, HuR KO1 did not form a tumor (Fig. 1j, k). We also knocked out HuR expression in SUM159 cells to investigate the generality of the above findings in MDA-MB-231 cells. Similarly, two SUM159 HuR KO clones had reduced protein expression of HuR targets (Supplementary Fig. 5a, b) and grew slower than parental or sgControl cells (Supplementary Fig. 5c). However, in contrast to MDA-MB-231 HuR KO clones, SUM159 HuR KO clones could form colonies, but the colony numbers were fewer than those formed by parental and sgControl cells (Supplementary Fig. 5d, e). SUM159 HuR KO clones invaded less well (Supplementary Fig. 5f, g) and had delayed tumor formation as well as decreased tumor size compared to sgControl cells (Supplementary Fig. 5h, i). These results indicate that HuR promotes breast cancer cell proliferation, invasion and tumorigenesis, and are consistent with previous findings in pancreatic and colon cancer cells[31]. These findings demonstrate that HuR plays a critical role in cell proliferation, invasion and tumorigenesis of breast cancer, making it a promising therapeutic target.

**KH-3 is a small molecule inhibitor of HuR.** After discovery of first candidate HuR inhibitors[17], we continued our efforts with the objective to identify inhibitors with increased potency. We screened 2000 additional compounds composed of 1673 compounds from the National Cancer Institute plus 291 in-house compounds using the fluorescence polarization (FP) assay described previously[17]. The same full-length HuR protein and a 16-mer ARE[Msi1] RNA oligomer from 3′-UTR of Musashi1 (Msi1, a known HuR target[32]) mRNA were used in the screening. Among the initial hits, besides compound Aza-9, for which we recently reported nuclear magnetic resonance (NMR) study of its binding to HuR[18], KH-3 was another top hit (Fig. 2a). Compound KH-3B, which is structurally similar to KH-3 (Fig. 2b), showed no

inhibition at the same concentration; so, it was used as a negative control of KH-3 in the later studies.

The effect of KH-3 disrupting HuR–ARE[Msi1] interaction was validated using FP and AlphaLISA assays. In both assays, KH-3 displayed dose-dependent inhibitory effect with a Ki of 0.83 μM and 0.72 μM, respectively. KH-3B only exhibited a minor effect at the highest dose tested (Fig. 2c, d). Surface plasmon resonance (SPR) was performed to verify the direct binding of KH-3 to HuR protein. KH-3 bound to full-length HuR protein (Fig. 2e) and shorter fragment RRM1/2 (Supplementary Fig. 6a) in a dose-dependent manner; whereas KH-3B showed slight binding at the highest concentration (Supplementary Fig. 6b, c). We applied computational docking using the reported crystal structure of HuR RRM1/2 in complex with RNA (PDB 4ED5) as a starting point[33]. Figure 2f represents the top-scoring computational models for each of KH-3 and KH-3B in complex with HuR RRM1/2. In both models, the ligand occupies a similar orientation in the RNA-binding groove, making use of similar interactions to the uracil base that engages this site in the crystal structure. In the KH-3 model, the hydroxamic acid moiety forms two hydrogen bonds to residues Ser-94 and Ile-133 of HuR RRM1/2 protein; in contrast, KH-3B does not have the hydroxamic acid group and cannot form an analogous hydrogen-bonding pattern. This in silico docking result provides a potential structural rationale for why KH-3 binds to HuR more tightly than KH-3B.

To confirm that KH-3 binds to endogenous HuR and disrupts HuR–mRNA interactions in cells, we employed cellular thermal shift assay (CETSA), RNA pull down and ribonucleoprotein immunoprecipitation (RNP IP) in MDA-MB-231 cells. KH-3 pre-treatment induced thermal stabilization of HuR compared to DMSO vehicle control (Fig. 2g, h and Supplementary Fig. 3), which demonstrates the direct binding of KH-3 to HuR in cells. In the pull-down experiment, KH-3, but not KH-3B, inhibited endogenous HuR from binding a biotinylated ARE[Msi1] RNA oligomer and being pulled down with streptavidin-coated beads (Fig. 2i, j and Supplementary Fig. 3), indicating the ability of KH-3 to block HuR–ARE interactions. In the RNP IP assay, the significant enrichment of three mRNAs immunoprecipitated by HuR antibody compared to IgG verified that these three genes are HuR direct targets in MDA-MB-231 cells. KH-3, but not KH-3B, significantly disrupted the interaction between HuR and three target mRNAs (Fig. 2k). This assay was also performed in SUM159 cells and similar result was obtained (Supplementary Fig. 6d).

**KH-3 blocks HuR function and inhibits breast cancer growth.** After validation of KH-3 as a competitive HuR inhibitor with various biochemical/biophysical assays, the effects of KH-3 on human breast cancer cells were examined. KH-3 exhibited potent cytotoxicity against a panel of TNBC cell lines with IC50 less than

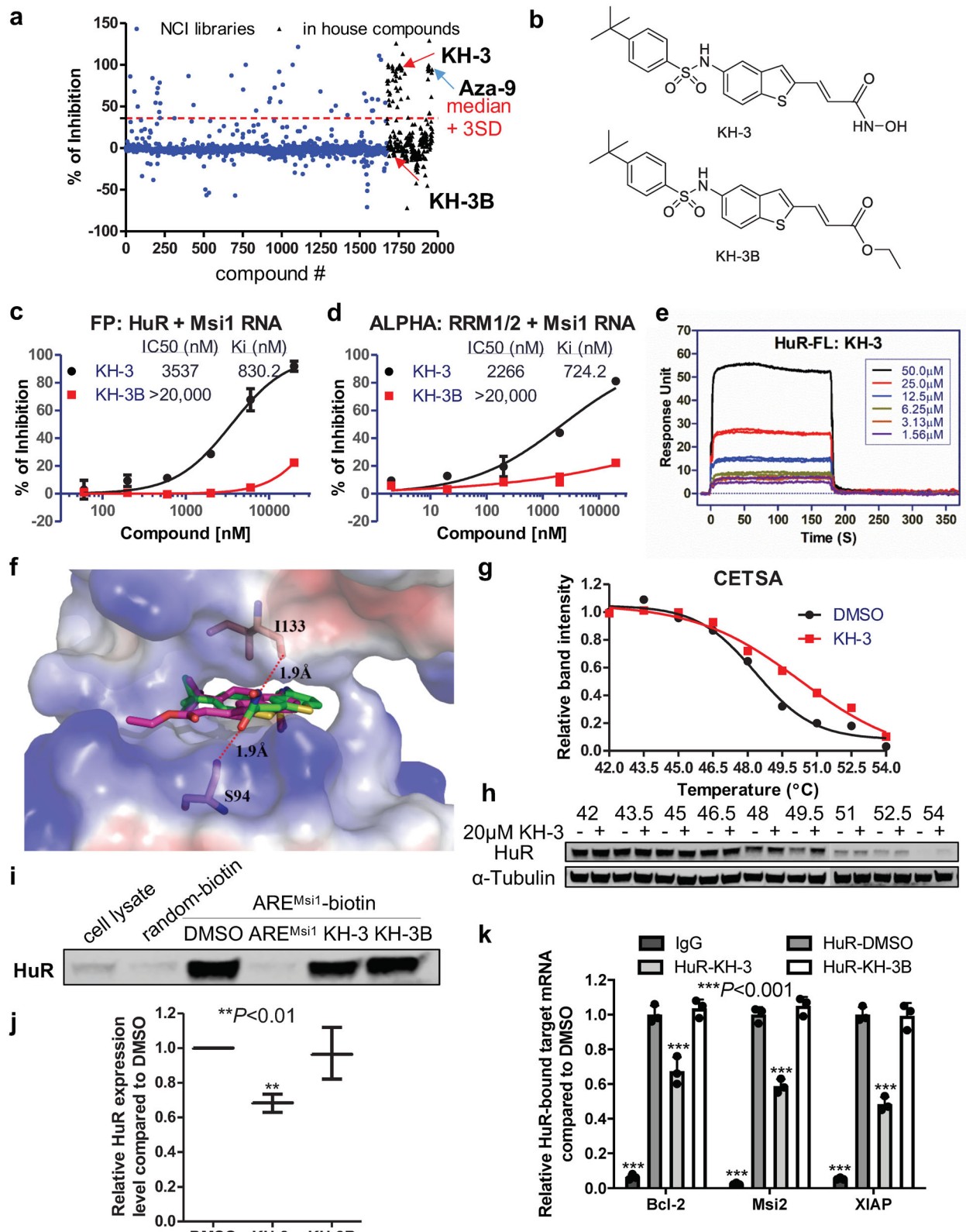

10 μM except for Hs578T cell line in MTT-based cytotoxicity assay (Fig. 3a). Compared to the parental cells and sgControl, two HuR KO clones were less sensitive to KH-3, with the $IC_{50}$ increased more than two-fold (from ca. 4 μM to 10–11 μM, Fig. 3b). This result indicates that HuR knockout attenuates the inhibitory effect of KH-3, and KH-3 cellular activity is HuR dependent.

Since HuR stabilizes and facilitates the translation of target mRNA, we next tested whether KH-3 could block these HuR functions by determining the mRNA half-life and the encoded protein expression levels in MDA-MB-231 and SUM159 cells. KH-3, but not its inactive analog KH-3B, decreased the half-life of Bcl-2, Mis2 and XIAP mRNA (Fig. 3c–h). Similarly, the protein expression levels of Bcl-2, Msi2 and XIAP were reduced by KH-3

**Fig. 2 Identification of KH-3 as an inhibitor of HuR–mRNA interaction. a** HTS in ~2000 compounds from the NCI libraries and in-house compounds using FP assay. Shown here is the scattergram of compound activity expressed as % of inhibition. Median + 3 SD was used as a threshold to pick initial hits. KH-3 and Aza-9 are two top hits. KH-3B has no activity in the screening. **b** Chemical structures of KH-3 and KH-3B. **c** Dose–response curve of KH-3 and negative control KH-3B disrupting HuR–ARE^Msi1 binding in FP assay using 10 nM HuR protein and 2 nM fluorescein-labeled Msi1 RNA ($n = 3$). **d** Dose–response curve of KH-3 and KH-3B disrupting HuR–ARE^Msi1 binding in ALPHA assay using 100 nM HuR RRM1/2 protein and 25 nM biotin-labeled Msi1 RNA ($n = 3$). **e** SPR analysis of KH-3 binding to immobilized full-length HuR protein. Six doses were used and duplicated. **f** Computational docking of KH-3 and KH-3B to HuR RRM1/2. The protein is shown in surface representation with positive charges in blue and negative charges in red. KH-3 (green) and KH-3B (magenta) are shown in sticks. **g** Cellular thermal shift curves of HuR in MDA-MB-231 cells treated with DMSO or 20 μM KH-3, values are mean from two independent experiments. **h** Representative western blot results from one experiment. **i**, **j** Pull-down analysis of KH-3 disrupting ARE^Msi1 oligo binding to endogenous HuR in MDA-MB-231 cells. Random oligo is used as a negative control of the assay and unlabeled ARE^Msi1 oligo is used as a positive control. **i** Representative western blot result from one experiment. **j** Quantified relative HuR expression. Values are mean ± SD from $n = 3$ independent experiments (**$P < 0.01$, one-way ANOVA). **k** RNP IP analysis of HuR bound mRNAs affected by KH-3 in MDA-MB-231 cells. Isotype IgG is used as a negative control of HuR antibody. Values are mean ± SD from $n = 3$ independent experiments (***$P < 0.001$, two-way ANOVA).

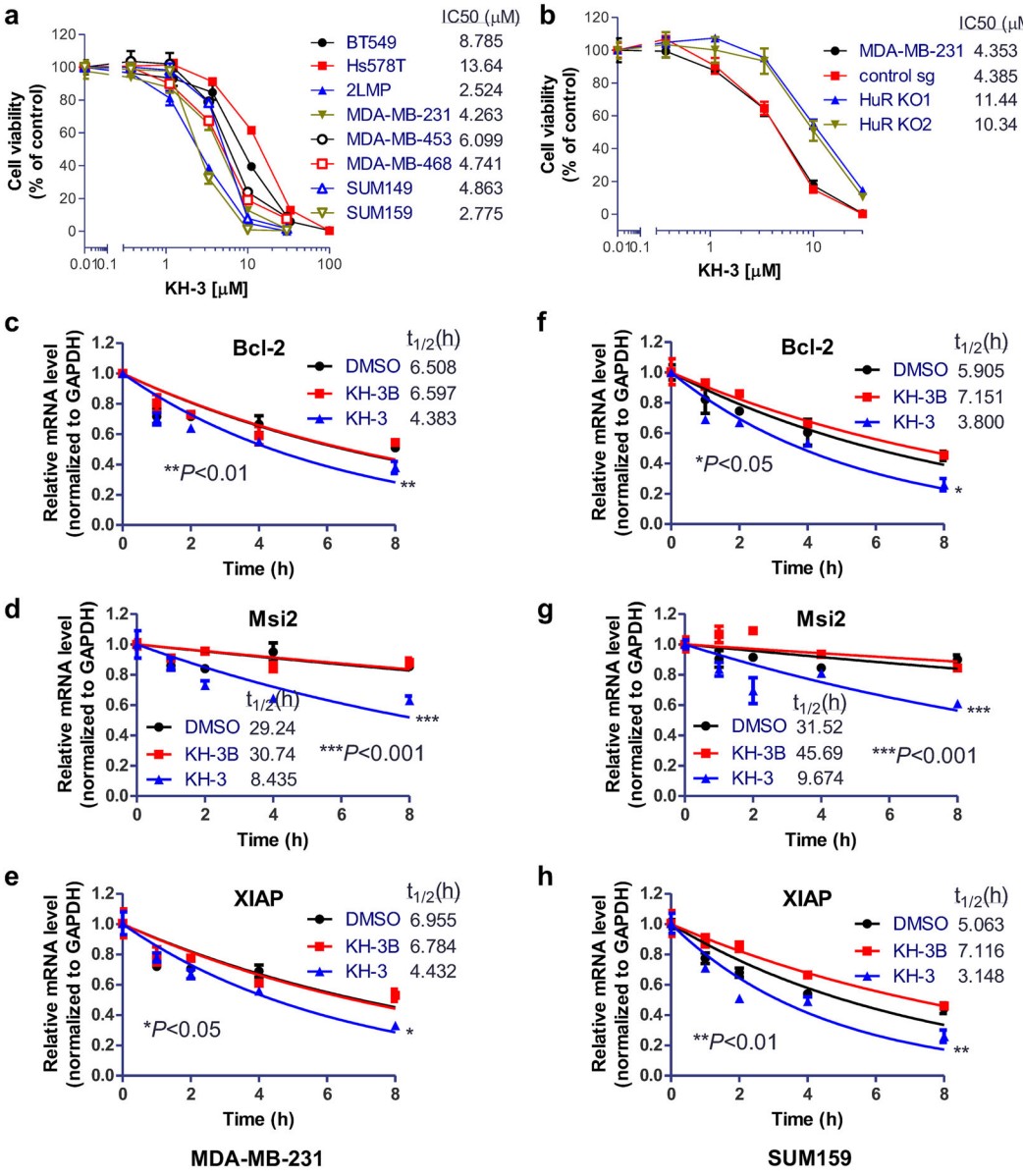

**Fig. 3 KH-3 inhibits breast cancer cell growth and destabilizes HuR targets. a**, **b** MTT-based cytotoxicity of KH-3 against a panel of TNBC cell lines (**a**) and MDA-MB-231 cells, sgControl and two HuR KO clones (**b**). **c–h** Half-life of Bcl-2, Msi2 and XIAP mRNA in MDA-MB-231 (**c–e**) and SUM159 (**f–h**) cells treated with 5 μg/mL actinomycin D together with DMSO, KH-3 or KH-3B. Data are mean ± SD from $n = 3$ independent experiments (*$P < 0.05$, **$P < 0.01$, ***$P < 0.001$, two-way ANOVA).

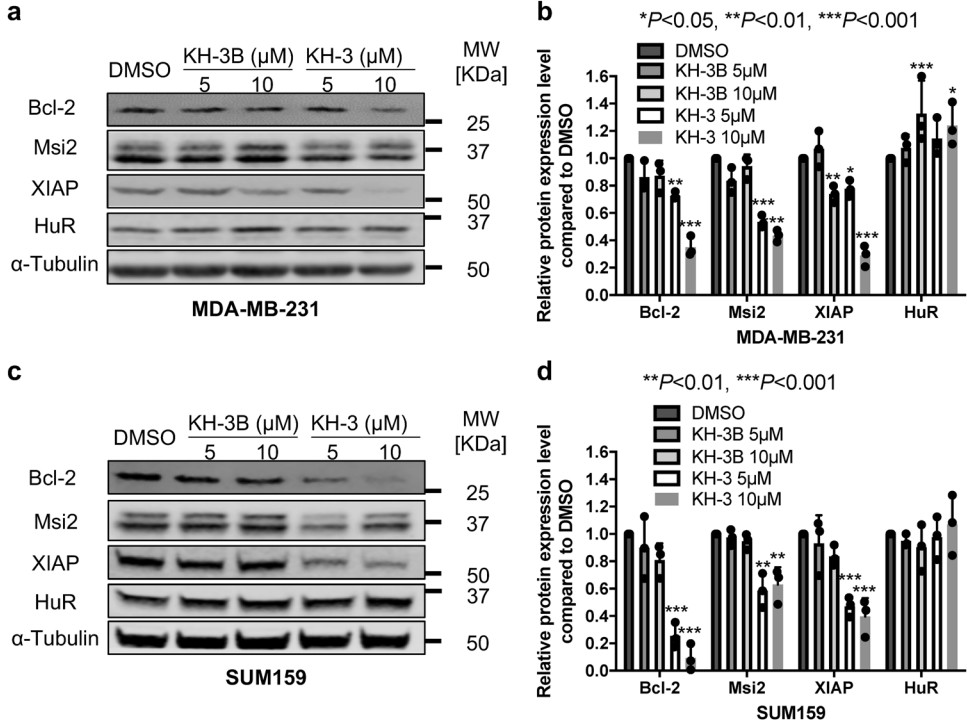

**Fig. 4 KH-3 decreases protein expression levels of HuR targets. a–d** Protein levels of Bcl-2, Msi2, XIAP, and HuR in MDA-MB-231 (**a**, **b**) and SUM159 (**c**, **d**) cells treated with DMSO, KH-3 or KH-3B at the indicated doses for 48 h. α-Tubulin is used as loading control. Representative western blot results from one experiment in MDA-MB-231 (**a**) and SUM159 (**c**) cells. Quantified relative expression of HuR and downstream target Bcl-2, Msi2 and XIAP in MDA-MB-231 (**b**) and SUM159 (**d**) cells. Values are mean ± SD from n = 3 independent experiments (*P < 0.05, **P < 0.01, ***P < 0.001, two-way ANOVA).

but not KH-3B as compared to DMSO (Fig. 4a–d and Supplementary Fig. 3). KH-3B decreased about 25% of XIAP protein level at 10 μM. Meanwhile, HuR protein levels were not decreased by KH-3 treatment, which indicates that KH-3 works through disruption of HuR–mRNA interactions without HuR protein degradation.

**KH-3 inhibits breast cancer cell migration and invasion.** To determine if KH-3 treatment yields similar effect on cell invasion to HuR depletion, we carried out scratch and invasion assays in MDA-MB-231 and SUM159 cells. In the scratch assay, the wound width was significantly wider in samples treated with KH-3 for 24 h compared to DMSO treatment, which indicates that cell migration is inhibited by KH-3 (Fig. 5a–c). In the invasion assay, KH-3 decreased invaded cell numbers in a dose-dependent manner while KH-3B had no effect (Fig. 5d–f).

We then explored the KH-3 mechanism of action in blocking cell migration and invasion using a qPCR array that consists of 88 human tumor invasion/metastasis genes. A heatmap view with hierarchical clustering revealed substantial similarities in the gene expression profile (Nine undetectable genes were excluded) for KH-3 treated cells to that for HuR KO1 clone (Fig. 5g). KH-3 treatment upregulated several metastatic suppressors (e.g., CD82 and CDH1)[34] and downregulated several genes frequently over-expressed in breast cancer lung metastasis (such as CDH2)[35]. The induction of CDH1 expression was verified by CDH1 promoter/ luciferase reporter assay. Compared to DMSO control, KH-3 increased the relative luciferase signals in a dose-dependent manner in cells transfected with a vector containing firefly luciferase gene driven by CDH1 promoter, but not in cells transfected with a control vector (Fig. 5h). As CDH1 encodes protein E-cadherin, which is an epithelial marker, the induction

of CDH1 may suggest that KH-3 inhibits cell migration and invasion through blockade of epithelial–mesenchymal transition.

**KH-3 inhibits the interaction between HuR and FOXQ1 mRNA.** Binding to the mRNA 3′-UTR and promoting the expression of Snail, which leads to the repression of CDH1 and subsequently its encoded protein E-cadherin, is one of the mechanisms that HuR is reportedly involved in cell migration[14]. However, in our study, Snail appears not to be a direct HuR target in either MDA-MB-231 or SUM159 (Supplementary Fig. 7a). Meanwhile, KH-3 treatment does not decrease, but increases, Snail mRNA levels (Supplementary Fig. 7b). So far we do not know how KH-3 upregulates CDH1 and whether this regulation is dependent on HuR. To address these questions and to identify direct HuR targets contributing to HuR's function in breast cancer invasion and metastasis, we employed two types of high-throughput analysis: RIP-seq and RNA-seq. In RIP-seq, anti-HuR antibody was used to immunoprecipitate endogenous HuR-RNA complexes in MDA-MB-231 cells and the HuR-bound RNAs were identified to obtain a list of direct HuR targets by comparing to that of isotype IgG control (Supplementary Data 1). RNA-seq was performed using mRNA samples collected from MDA-MB-231 cells treated with KH-3 and the result disclosed a list of genes affected by KH-3 treatment by comparing to that of DMSO control (Supplementary Data 2). Gene set enrichment analysis (GSEA) of epithelial-mesenchymal transition hallmark set for RNA-seq data exhibited that EMT genes were enriched in DMSO group compared to KH-3 group (Supplementary Fig. 7c), indicating that KH-3 treatment inhibits EMT signature. Intersecting the two lists from RIP-seq and RNA-seq revealed a set of 995 common targets (Fig. 6a). Among them, FOXQ1, a transcription factor that belongs to Forkhead box protein family, is one of the top genes inhibited by KH-3. FOXQ1 has been reported to be

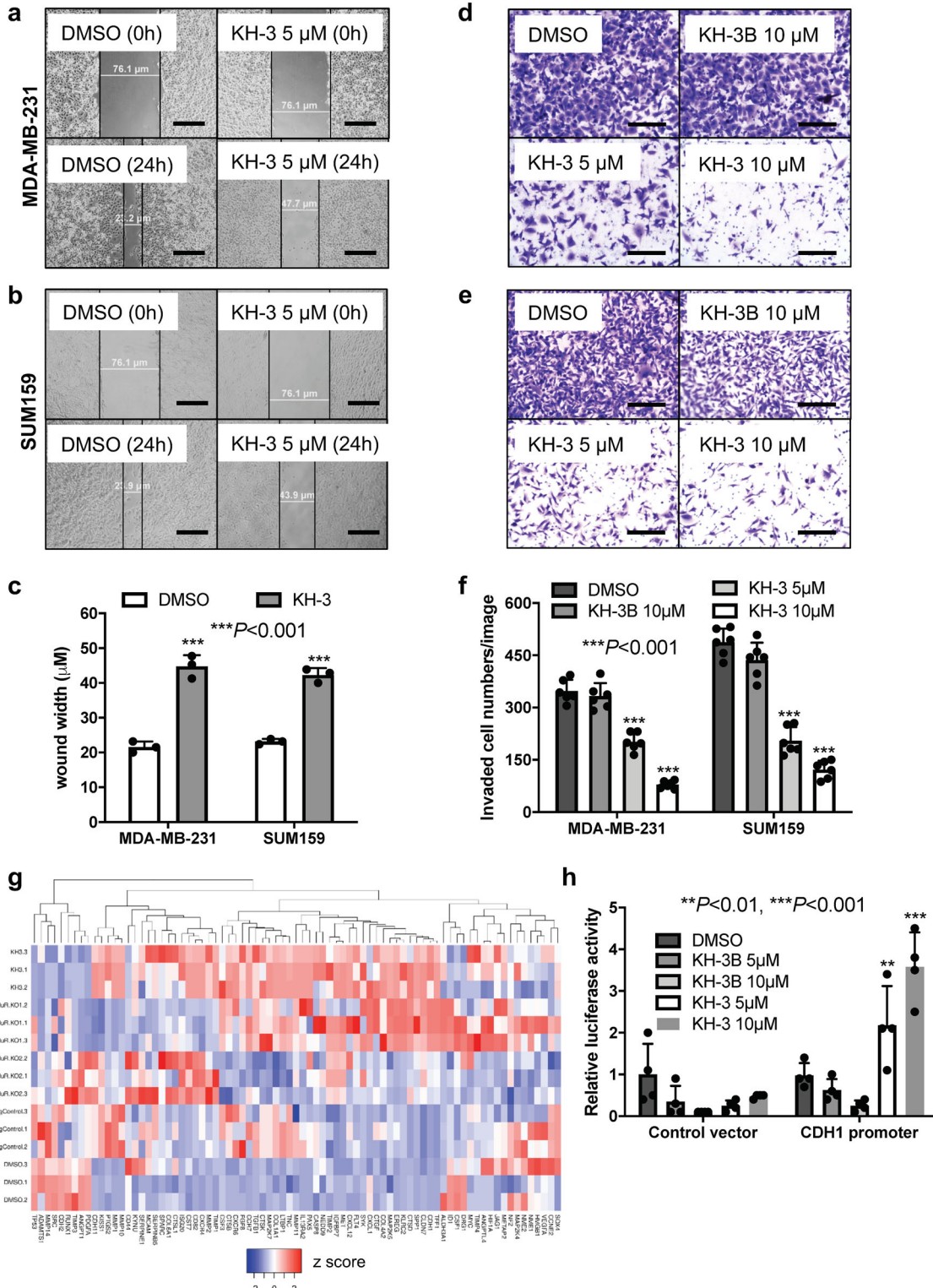

**Fig. 5 KH-3 inhibits breast cancer cell migration and invasion and promotes CDH1 expression. a–c** Scratch assay in MDA-MB-231 and SUM159 cells treated with DMSO or KH-3. **a, b** Representative images of cell migration at 0 and 24 h after scratching with indicated treatment in MDA-MB-231 (**a**) and SUM159 (**b**) cells, scale bars: 50 μm. **c** Wound widths in two cell lines 24 h after scratching and treatment (***$P < 0.001$, $t$-test, n = 3). **d–f** Invasion assay in MDA-MB-231 and SUM159 cells treated by DMSO, KH-3B or KH-3. **d, e** Representative images of stained invaded cells with indicated treatment in MDA-MB-231 (**d**) and SUM159 (**e**) cells, scale bars: 200 μm. **f** Invaded cell numbers per image in both cell lines with indicated treatment (***$P < 0.001$, one-way ANOVA, $n = 6$). **g** The heatmap view of PCR pathway array focusing on invasion and metastasis related genes. The relative mRNA levels were presented as z score, each treatment was triplicated. **h** CDH1 luciferase reporter assay in HEK 293FT cells treated by DMSO, KH-3B or KH-3. Values are mean ± SD from $n = 4$ independent experiments (**$P < 0.01$, ***$P < 0.001$, two-way ANOVA).

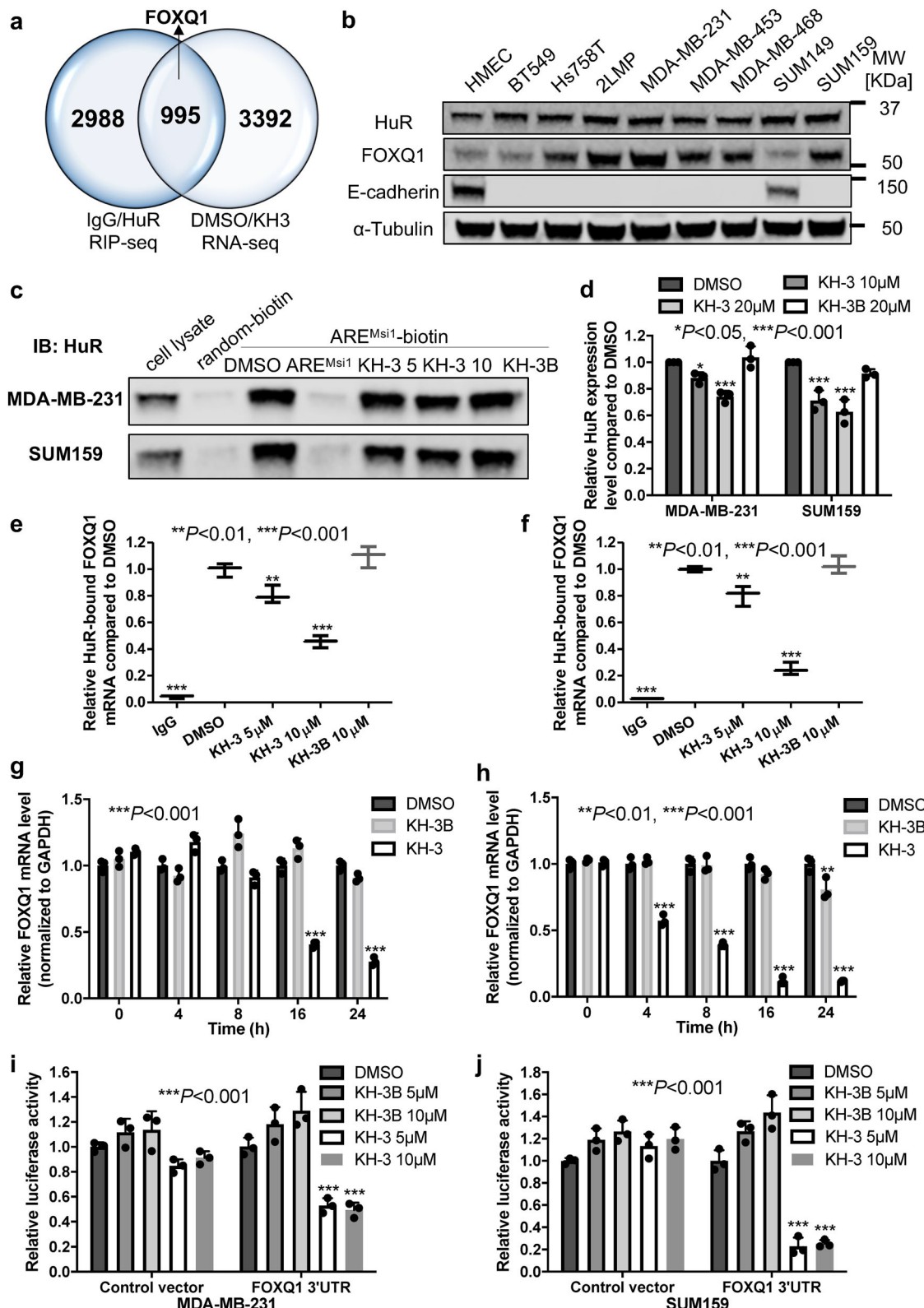

implicated in breast cancer EMT and metastasis, negative regulation of *CDH1* expression is one of the mechanisms[36]. FOXQ1 protein level is upregulated in MDA-MB-231, its subclone 2LMP, MDA-MB-453, MDA-MB-468 and SUM159 cells compared to Human Mammary Epithelial Cells (HMEC). Reversely, the expression of E-cadherin is lost in those cells with high FOXQ1 (Fig. 6b and Supplementary Fig. 3). This result suggests a negative

correlation between FOXQ1 and E-cadherin expression. There is no obvious correlation between FOXQ1 and total HuR expression among different breast cancer cell lines, since total HuR is expressed in all breast cancer cell lines tested.

We validated FOXQ1 as a direct HuR target using RNA pull down and RNP IP assays in MDA-MB-231 and SUM159 cells. Meanwhile, we examined whether KH-3 could disrupt the

**Fig. 6 FOXQ1 is a HuR target and KH-3 disrupts HuR–FOXQ1 interaction. a** Venn diagram depicting the number of targets identified in two independent RNA-seq experiments. FOXQ1 is a direct HuR target, which is also one of the top mRNAs decreased by KH-3 treatment. **b** Protein expression levels of HuR, FOXQ1 and E-cadherin in HMEC and a panel of TNBC cell lines. **c, d** Pull-down analysis of KH-3 disrupting ARE$^{FOXQ1}$ oligo binding to endogenous HuR in MDA-MB-231 and SUM159 cells. **c** Representative western blot result from one experiment. **d** Quantified relative HuR expression. Values are mean ± SD from $n = 3$ independent experiments (*$P < 0.05$, ***$P < 0.001$, one-way ANOVA). **e, f** RNP IP analysis of HuR bound FOXQ1 mRNA affected by KH-3 in MDA-MB-231 (**e**) and SUM159 (**f**) cells. Values are mean ± SD from three independent experiments (**$P < 0.01$, ***$P < 0.001$, one-way ANOVA). **g, h** Relative FOXQ1 mRNA levels in MDA-MB-231 (**g**) and SUM159 (**h**) cells treated with DMSO, KH-3 or KH-3B at the indicated time points. Values are mean ± SD from $n = 3$ independent experiments (***$P < 0.001$, two-way ANOVA). **i, j** FOXQ1 3′-UTR luciferase reporter assay in MDA-MB-231 (**i**) and SUM159 (**j**) cells treated by DMSO, KH-3B or KH-3. Values are mean ± SD from $n = 3$ independent experiments (***$P < 0.001$, two-way ANOVA).

HuR–FOXQ1 mRNA interaction. We designed a biotinylated 18-mer ARE$^{FOXQ1}$ RNA oligomer from 3′-UTR of FOXQ1 mRNA to pull down endogenous HuR. The biotinylated ARE$^{FOXQ1}$ RNA oligomer, but not oligomer with random sequence, could considerably pull down HuR protein, whereas the 10-fold excess of unlabeled ARE$^{FOXQ1}$ RNA oligomer completely abolished the HuR protein pulled down, indicating a strong HuR–ARE$^{FOXQ1}$ interaction (Fig. 6c, d). KH-3, but not KH-3B, decreased the amount of HuR protein pulled down by biotinylated ARE$^{FOXQ1}$ RNA oligomer, demonstrating the disruption of HuR–ARE$^{FOXQ1}$ interaction by KH-3 (Fig. 6c, d). In the RNP IP assay, compared to isotype IgG, the marked enrichment of FOXQ1 mRNA immunoprecipitated by HuR antibody confirmed the binding of HuR to FOXQ1 mRNA. KH-3, but not KH-3B, disrupted the HuR–FOXQ1 mRNA interaction in a dose-dependent manner (Fig. 6e, f).

We also investigated if KH-3 could destabilize FOXQ1 mRNA. However, there were minor differences in mRNA half-lives among different treatment groups (Supplementary Fig. 8a, b). It is possibly because the half-life of FOXQ1 mRNA is very short (ca. 1 h, Supplementary Fig. 8a, b) and KH-3 is not quick enough to act. Therefore, we checked FOXQ1 mRNA at different time points after treatment. Compared with that of DMSO, KH-3 decreased FOXQ1 mRNA levels in a time-dependent manner (Fig. 6g, h). However, the mRNA level was only reduced 20% at 8 h in MDA-MB-231 and about 40% at 4 h in SUM159, which means it takes longer time for KH-3 to act than the half-life of FOXQ1 mRNA. KH-3B caused about 20% reduction of FOXQ1 mRNA at 24 h in SUM159.

We also determined FOXQ1 mRNA levels in HuR KO clones treated with KH-3 to explore whether the reduction of FOXQ1 mRNA level by KH-3 is dependent on HuR or not. As shown in Supplementary Fig. 8c, although KH-3 treatment still decreased FOXQ1 mRNA level in two HuR KO clones, the reduction was significantly attenuated compared to that in sgControl cells, indicating that the inhibitory effect of KH-3 on FOXQ1 was at least partially dependent on HuR.

Given HuR mainly binds to the 3′-UTR of target mRNA, we verified the inhibitory effect of KH-3 on FOXQ1 3′-UTR using 3′-UTR reporter assay. KH-3, but not KH-3B, reduced the luciferase signal in cells transfected with vector bearing FOXQ1 3′-UTR, while it did not have significant effect on cells with control vector in MDA-MB-231 and SUM159 cells (Fig. 6i, j), indicating a specific inhibition of FOXQ1 3′-UTR.

**FOXQ1 contributes to HuR's role in promoting cell invasion**. We sought to determine if FOXQ1 contributes to HuR's functional role in promoting breast cancer cell invasion by induction of exogenous FOXQ1. As shown in Fig. 7a, b, compared to sgControl cells transfected with control vector, HuR KO clones transfected with control vector had fewer invaded cells. However, exogenous FOXQ1 partially rescued invasive capability impaired by HuR knockout as the invaded cell number increased in HuR KO clones transfected with FOXQ1 cDNA compared to those

transfected with control vector, although the invaded cell numbers were still fewer than sgControl cells. This result indicates that FOXQ1 is one of the downstream targets that contribute to HuR's role in cell invasion. Overexpression of FOXQ1 was verified (Fig. 7c). We also determined the mRNA levels of CDH1 and CD82 (Fig. 7c), two genes upregulated by HuR KO and KH-3 in qPCR array. HuR KO clones transfected with control vector had significantly higher CDH1 mRNA levels compared to sgControl cells transfected with control vector. Overexpression of FOXQ1 abolished the induction CDH1 expression in both HuR KO clones, but HuR KO1 still had higher CDH1 than basal level of sgControl while the level of CDH1 in HuR KO2 was not significantly different to basal level. CD82 mRNA levels was significantly increased in HuR KO2 clone compared to sgControl cells. Exogenous FOXQ1 did not affect CD82 mRNA level in sgControl cells and HuR KO clones. These data suggest that FOXQ1 contribute to HuR's role in cell invasion through downregulating CDH1.

We then tested if FOXQ1 overexpression could also rescue cell invasive capability inhibited by KH-3. As shown in Fig. 7d, e, compared to DMSO, KH-3 significantly inhibited invaded cell numbers in MDA-MB-231 cells transfected with control vector. However, exogenous induction of FOXQ1 eliminated this effect as KH-3 treatment did not decrease invaded cell numbers in cells transfected with FOXQ1 cDNA, demonstrating that KH-3 inhibits cell invasion through suppression of FOXQ1 expression. Similarly, we checked CDH1 and CD82 mRNA levels (Fig. 7f). KH-3 markedly induced CDH1 expression in cells transfected with control vector while this induction was recalled by overexpression of FOXQ1. FOXQ1 overexpression had no effect on KH-3 induced CD82 upregulation. We also detected the protein expression levels of HuR targets (Fig. 7g, h and Supplementary Fig. 3). KH-3 decreased FOXQ1 protein level, which was abolished by FOXQ1 overexpression. KH-3 also reduced protein levels of other HuR targets such as Bcl-2, Msi2 and β-catenin, none of which were significantly affected by FOXQ1 overexpression.

**KH-3 inhibits breast cancer growth and metastasis in vivo**. The anti-tumor efficacy of KH-3 was examined in MDA-MB-231 orthotopic xenograft model. Tumor-bearing female athymic mice were randomized into two groups and treated with either vehicle control or 100 mg/kg KH-3, via intraperitoneal injection three times/week for three weeks. KH-3 significantly inhibited tumor growth, resulting in 60% tumor regression after three-week treatment (Fig. 8a). KH-3 reduced the protein expression levels of HuR targets in tumor tissues as well as the induction of E-cadherin expression (Fig. 8b and Supplementary Fig. 3).

The anti-metastatic potential of KH-3 was then explored in an experimental metastasis model by injecting luciferase-expressing 2LMP cells into the lateral tail vein of athymic mice. The mice were randomized into two groups and treated with either vehicle or KH-3 via intraperitoneal injection five times/week for five weeks. We monitored the tumor initiation and progression in

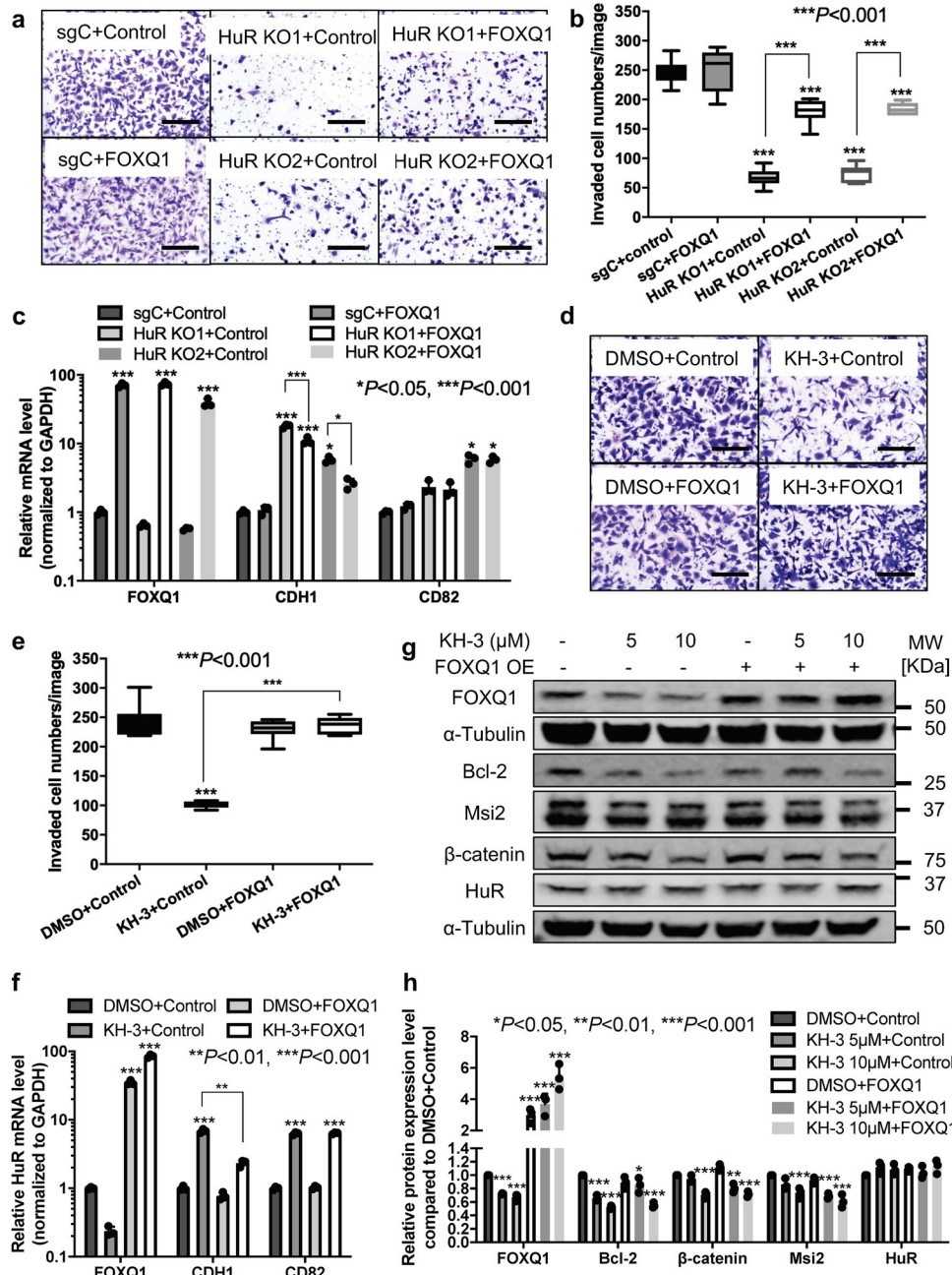

**Fig. 7 FOXQ1 contributes to HuR role in promoting cell invasion. a**, **b** Invasion assay in parental MDA-MB-231 cells, sgControl and two HuR KO clones transfected with control vector or vector containing FOXQ1 cDNA. **a** Representative images of stained invaded cells, scale bars: 200 μm. **b** The number of invaded cells per image (***$P < 0.001$, one-way ANOVA, $n = 6$). **c** mRNA expression levels of FOXQ1, CDH1 and CD82 in parental MDA-MB-231 cells, sgControl and two HuR KO clones transfected with control vector or vector containing FOXQ1 cDNA. Values are mean ± SD from $n = 3$ independent experiments (*$P < 0.05$, ***$P < 0.001$, two-way ANOVA). **d**, **e** Invasion assay in MDA-MB-231 cells transfected with control vector or vector containing FOXQ1 cDNA together with DMSO or 10 μM KH-3 treatment. **d** Representative images of stained invaded cells, scale bars: 200 μm. **e** The number of invaded cells per image (***$P < 0.001$, one-way ANOVA, $n = 6$). **f** mRNA expression levels of FOXQ1, CDH1 and CD82 in MDA-MB-231 cells transfected with control vector or vector containing FOXQ1 cDNA together with treatment of DMSO or 10 μM KH-3. Values are mean ± SD from $n = 3$ independent experiments (**$P < 0.01$, ***$P < 0.001$, two-way ANOVA). **g**, **h** Protein expression levels of FOXQ1, Bcl-2, Msi2, β-catenin, and HuR in MDA-MB-231 cells transfected with control vector or vector containing FOXQ1 cDNA together with treatment of DMSO or KH-3 at the indicated doses for 48 h. α-Tubulin is used as loading control. **g** Representative western blot results from one experiment. **h** Quantified relative expression of HuR and downstream targets. Values are mean ± SD from $n = 3$ independent experiments (*$P < 0.05$, **$P < 0.01$, ***$P < 0.001$, two-way ANOVA).

lung by bioluminescent imaging as well as the survival. Supplementary Fig. 9a presented representative images for the same mice at three stages: (I) mouse 3 with initial detection of pulmonary metastasis; (II) mouse 1 with initial detection of pulmonary metastasis and mouse 3 with pulmonary metastasis progression; (III) mouse 1 with pulmonary metastasis progression and near moribund mouse 3 with extensive pulmonary metastases. KH-3 treatment significantly delayed the initiation of pulmonary metastases (Fig. 8c). The median time for the two groups was 38 and 71 days, respectively. KH-3 also decreased the

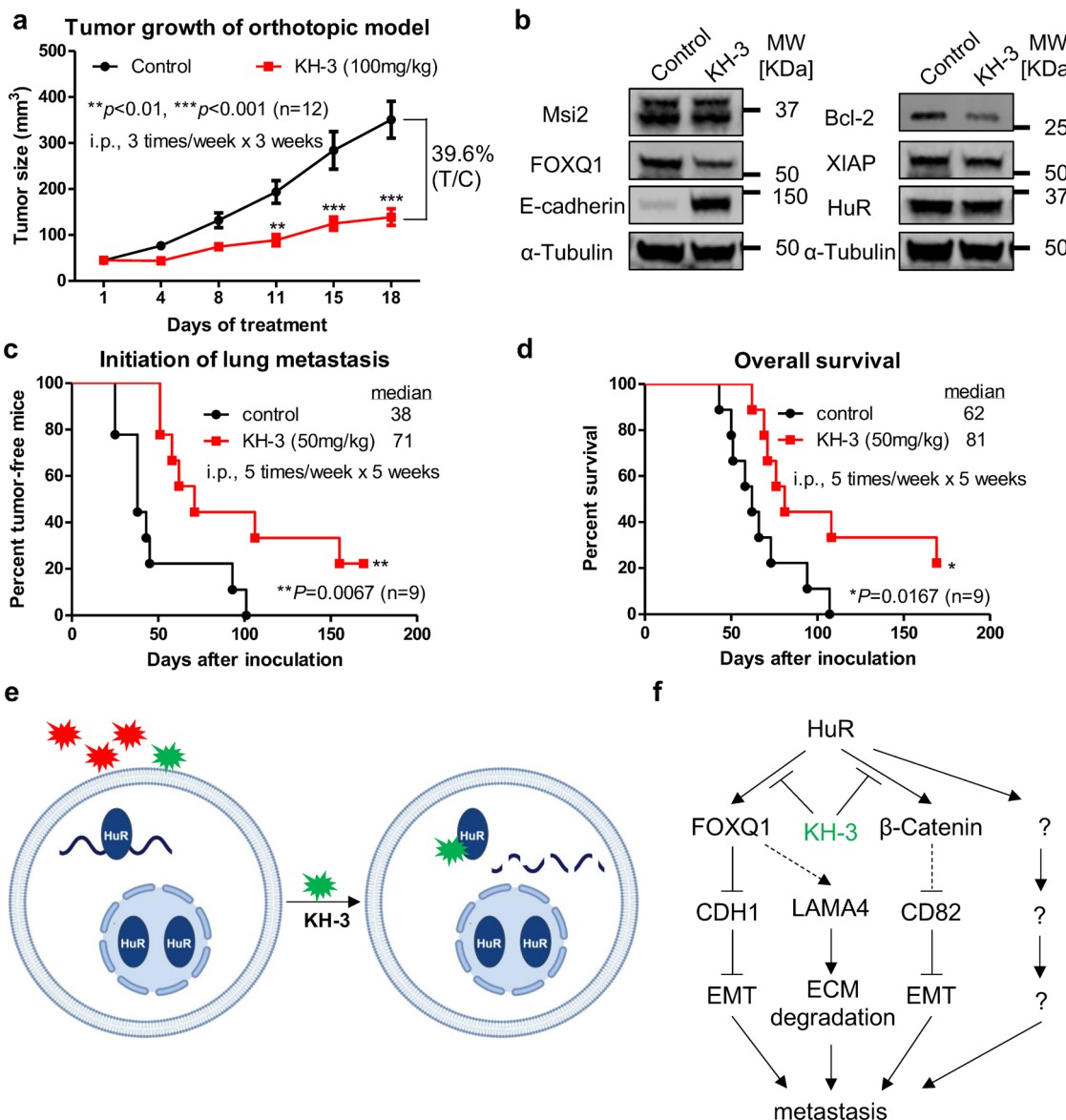

**Fig. 8 KH-3 inhibits breast cancer growth and metastasis in vivo. a** In vivo anti-tumor efficacy of KH-3 in MDA-MB-231 orthotopic xenograft model. KH-3 treatment significantly inhibits the tumor growth compared to the vehicle control ($n = 12$, **$P < 0.01$, ***$P < 0.001$, two-way ANOVA). **b** Protein expression levels of HuR and downstream targets in tumor tissues after treatment. α-Tubulin is used as loading control. **c, d** Kaplan–Meier analysis of the initiation of pulmonary metastases (**c**) and overall survival of mice (**d**) comparing treatment with vehicle control or KH-3 in an experimental metastasis model ($n = 9$, *$P < 0.05$, **$P < 0.01$, log-rank test). **e** Schematic of screening for HuR inhibitors that bind to HuR and block HuR function. **f** Proposed working model of KH-3 inhibiting breast cancer invasion and metastasis.

metastasis incidence. All mice (9/9) in control group had pulmonary metastases while 77.7% (7/9) mice in KH-3 group had pulmonary metastases at the end of experiment. KH-3 treatment significantly improved the survival time of mice as well. The median survival time in control group was 62 days while 81 days in KH-3 group (Fig. 8d). At the conclusion of the experiment, all lung tissues were collected and performed immunohistochemistry staining. Supplementary Fig. 9b presented representative IHC H&E staining images of lungs, which displayed tumor cells surrounded by lung cells. Besides the primary outcome, we also monitored side effects of KH-3. KH-3 treatment caused minor diarrhea in some mice. Some mice had swollen abdomens starting the fourth week of treatment, which may be induced peritonitis due to repeated intraperitoneal injection. No other side effects were noticed. The mice in KH-3 group gained weight similar to those in control group during the

first 43 days of the experiment (Supplementary Fig. 9c). After that, the mice in control group started to die so the weight curve could not be plotted. These data suggest that KH-3 is a potent and safe agent to inhibit breast cancer metastasis in vivo.

We also investigated the anti-metastatic potential of KH-3 in a 4T1 experimental metastatic model by injecting luciferase-expressing 4T1 cells into the lateral tail vein of BALB/c mice. The treatment dose and schedule were the same as used in 2LMP model. We used bioluminescence imaging to monitor the tumor initiation and progression. However, the imaging results of some mice were not consistent with the necropsy results obtained at the end of experiment (many lung tumor nodules were not luminescent). Therefore, the effect of KH-3 on tumor initiation was unable to be determined. Nevertheless, overall survival of mice was evaluated. Supplementary Fig. 9d presented the survival curves of mice in two groups. KH-3 treatment improved the

survival time of mice. The median survival time in control group was 36.5 days and in KH-3 group was 52.5 days.

## Discussion

In the current study, we find that high cytoplasmic HuR expression correlates with advanced tumor grade of breast cancer and poor overall and distant disease-free survival of breast cancer patients. Knocking out HuR in breast cancer cell lines not only decreases cells growth but also impairs cell invasion capability. These findings demonstrate that HuR is a promising therapeutic target for inhibiting breast cancer invasion and metastasis as well as tumor progression.

During the last decade, we and others reported the identification of small molecules that interfere HuR–RNA interactions through high-throughput screening[23] and one group published the structure-based design and optimization of an initial top hit[22]. Here, we identify a inhibitor KH-3 that disrupts HuR–mRNA interactions through competitively binding to HuR and therefore blocks HuR functions, leading to the decay of HuR target mRNA and/ or reduction of translation (Fig. 8e, f). As a result, KH-3 inhibits breast cancer cell growth and invasion in vitro and prohibits tumor growth and experimental lung metastasis in vivo.

HuR was reported to bind to 3′-UTR of Snail mRNA and promote its expression, which is one of the mechanisms that HuR is implicated in cell migration[14]. However, Snail is not a direct HuR target in the two breast cancer cell lines we tested. By intersecting the results from RIP-seq and RNA-seq and the follow-up validation, we find another transcriptional factor, FOXQ1, as a direct target of HuR in MDA-MB-231 and SUM159 cells, and that KH-3 can disrupt the interaction between HuR and FOXQ1 mRNA. Recently, a number of studies revealed the implication of FOXQ1 in cancer progression and metastasis[36–38]. Similar to Snail, FOXQ1 negatively regulates CDH1, by directly binding to the promoter region of CDH1[36]. In our study, FOXQ1 overexpression partially rescues upregulated CDH1 expression induced by HuR CRISPR KO and cell invasive capability prohibited by HuR CRISPR KO. Therefore, FOXQ1 is one of the HuR direct mRNA targets that contributes to HuR's role in breast cancer invasion and metastasis, potentially through promoting epithelial-to-mesenchymal transition. FOXQ1 was also reported to positively regulate several other genes involved in cancer invasion and metastasis, such as Snail[39] and LAMA4[37]. LAMA4, which encodes a secreted ECM protein, is found to promote tumor re-initiation in multiple organ microenvironments[37]. KH-3 treatment also reduces LAMA4 expression (Supplementary Fig. 10). Hence, FOXQ1 may also contribute to HuR's role in breast cancer invasion and metastasis via increasing the expression of protein that degrades ECM. However, more experiments are needed to support this hypothesis.

In qPCR array analyses, besides promoting the expression of CDH1, KH-3 treatment also induces the expression of CD82, a tumor metastasis suppressor. It is reported that transcription of CD82 can be downregulated by β-catenin (CTNNB1)[40]. β-catenin was found as a direct HuR target in MDA-MB-231[41], also verified in our study (β-catenin is one of the 995 overlaid genes of RIP-seq and RNA-seq, Fig. 6a). As FOXQ1 overexpression has no effect on KH-3 induced upregulation of CD82 and inhibition of β-catenin (Fig. 7f–h), we propose a working model that KH-3 disrupts HuR–FOXQ1 and HuR-β-catenin signaling axes in parallel, leading to the inhibition of invasion and metastasis (Fig. 8e, f). However, the role of HuR-β-catenin axis in breast cancer invasion and metastasis needs to be confirmed, and whether HuR inhibition induced CD82 upregulation is related to β-catenin reduction, remains further studies. HuR may also implicate in breast cancer invasion and metastasis through other

HuR–mRNA axes not tested in current study and these axes may or may not be interfered by KH-3.

As a small molecule inhibitor of HuR, KH-3 phenocopies HuR knockout in most of assays but less efficiently. KH-3 treatment results in ~60% tumor growth regression in xenograft model while HuR KO reduce tumor growth more than 95%. The anti-tumor efficacy of MS-444 and DHTS, two reported HuR inhibitors, were evaluated in colorectal cancer xenograft models. Treatment of HCT116 xenografts with 25 mg/kg MS-444, which blocks HuR function by interfering HuR cytoplasmic localization, results in about 1.7-fold reduction in tumor size[26]. Treatment of HCT116 xenografts with 10 mg/kg DHTS, which blocks HuR function by interfering the binding of HuR to RNA, results in ~4-fold reduction in tumor size[27]. The in vivo efficacy of three HuR inhibitors strongly supports that targeting HuR is a promising therapeutic strategy. However, like most molecularly targeted therapeutics, KH-3 may need to be combined with conventional chemotherapy regimens to maximize efficacy for inhibiting tumor progression and metastasis.

It was reported that DNA damaging agents could induce cytoplasmic translocation of HuR in pancreatic cancer cells[42]. Our preliminary studies show that docetaxel, a first-line chemotherapy of TNBC, also leads to cytoplasmic HuR accumulation and subsequently upregulation of HuR downstream targets. Likewise, HuR also promotes the translation of several target mRNA that encode proteins involved in cancer treatment resistance[29,43,44]. Hence, combinations of HuR inhibitor with chemotherapy have high chance to display synergistic effect and overcome the acquired chemoresistance, which will be further explored in our development of HuR inhibitors.

In conclusion, the small molecule inhibitor identified here can be used as a potent chemical probe to delineate HuR's functional roles and as a lead compound to develop novel molecular therapeutics targeting HuR. This study provides the therapeutic potential of HuR inhibition for lethal metastatic breast cancer with HuR overexpression.

## Methods

**Cell culture and reagents.** Human TNBC cell line BT-549, Hs 578T, MDA-MB-231, MDA-MB-453, and MDA-MB-468 as well as mouse TNBC cell line 4T1 were purchased from American Type Culture Collection (ATCC) and the 2LMP sub-clone which was generated from MDA-MB-231 formed lung metastasis in mice was a kind gift from Dr. Marc Lippman. Human embryonic kidney (HEK) 293FT cell line was purchased from ThermoFisher Scientific. These cell lines were cultured in DMEM (Mediatech) supplemented with 10% fetal bovine serum (FBS; Sigma-Aldrich), 1% antibiotics (Mediatech). Human TNBC cell line SUM-149 and SUM-159 were obtained from Asterand[45] and maintained in Ham's F12 (Mediatech) with 5% FBS, 10 μg/mL insulin and 500 ng/mL hydrocortisone (all from Sigma-Aldrich). Human Mammary Epithelial Cells (HMECs) were purchased from ATCC and cultured in Mammary Epithelial Cell complete medium (basal medium plus growth kit). All cell lines were either recently obtained or monitored by STR profiling.

(E)-3-(5-((4-(tert-Butyl)phenyl)sulfonamido)benzo[b]thiophen-2-yl)-N-hydroxyacrylamide (KH-3) was synthesized from (E)-3-(5-((4-(tert-butyl)phenyl)sulfonamido)benzo[b]thiophen-2-yl)acrylate (KH-3B) followed a literature[46]. To a solution of ethyl (E)-3-(5-((4-(tert-butyl)phenyl)sulfonamido)benzo[b]thiophen-2-yl)acrylate[47] (KH-3B, 4.0 g, 9.02 mmol) in methanol (20 mL) was added dropwise DBU (4.1 mL, 27.1 mmol). After 10 min, 50% aq hydroxylamine (5.5 mL) was added dropwise and the reaction mixture was allowed to stir at RT for 1 h. The reaction mixture was concentrated and the crude residue purified via silica gel MPLC (100% DCM to 10% MeOH in DCM) to afford an impure product, which was further purified by reverse phase HPLC (10% methanol in water to 100% methanol) to give the title compound as a light yellow solid (KH-3, 3.70 g, 95% yield). $^1$H NMR (400 MHz, DMSO-$d_6$): 7.79 (d, $J = 8.8$ Hz, 1 H), 7.71–7.54 (complex, 7 H), 7.16 (m, 1 H), 6.24 (d, $J = 15.4$ Hz, 1 H), 1.23 (s, 9 H); $^{13}$C NMR (100 MHz, DMSO-$d_6$): 161.9, 155.9, 141.2, 140.2, 136.9, 135.6, 134.3, 131.8, 127.8, 126.6, 126.2, 123.4, 120.8, 119.5, 114.5, 48.7, 30.8; purity (UPLC, 254 nm): >96%. KH-3 and KH-3B powder were dissolved in DMSO at 20 mM as stock solutions for in vitro assays. KH-3 powder was dissolved in PBS with 5% ethanol and 5% Tween-80 for animal studies.

The pCMV-XL5 vector with and without human FOXQ1 cDNA were obtained from OriGene. The pEZX-MT06 reporter vector with and without human FOXQ1

3′-UTR were obtained from GeneCopoeia. Antibodies against Bcl-2 (Cat# 2872, 1:500 dilution) and E-cadherin (Cat# 3195, 1:1000 dilution) were purchased from Cell Signaling Technology, HuR (3A2, Cat# sc-5261, 1:500 dilution), FOXQ1 (H-11, Cat# sc-166264, 1:500 dilution) and β-catenin (E-5, Cat# sc-7963, 1:500 dilution) were purchased from Santa Cruz Biotechnology. Anti-Msi2 (Cat# Ab76148, 1:2000 dilution) antibody was obtained from Abcam, Anti-XIAP (Cat# 610717, 1:500 dilution) antibody was obtained from BD Biosciences and anti-α-Tubulin (Cat# T5168, 1:4000 dilution) was obtained from Sigma-Aldrich. Anti-mouse (Cat# 926-68070 and Cat# 926-32210) and anti-rabbit (Cat# 926-68071 and Cat# 926-32211) IRDye secondary antibodies (1:15000 dilution) were purchased from LI-COR Biosciences.

The cell growth, MTT-based cytotoxicity assay, colony formation, western blot and qPCR (primers listed in Supplementary Table 2) were performed according to our previous publications[48,49]. The protein expression and purification, AlphaLISA assay, SPR, computational docking and mRNA stability assay were carried out as we previously described[17,18]. The intensities of immunoblots were quantified using Image Studio Ver 4.0 (LI-COR Biosciences) and normalized to loading control α-Tubulin.

**Patients and tissue microarray.** The study cohort consisted of 140 patients diagnosed with breast cancer who underwent primary surgical intervention between January 2005 and September 2012. Among them, 139 patients' survival information with up to 132 months' postoperative follow-up was documented. Patients' clinicopathologic variables, such as age, tumor grade and type, TNM stage, AJCC stage, survival time, recurrence time and metastasis were obtained from the medical records (Supplementary Table 1). The informed consent was obtained from these 140 patients. This study was reviewed and approved by the Institutional Review Board of the Air Force Medical University (Xi'an, China). A tissue microarray (TMA) constructed from formalin-fixed, paraffin-embedded tissue blocks from the above 140 patients was purchased from Shanghai Outdo Biotech Company (Shanghai, China). All procedures of construction TMA were performed as previously described[50].

**Immunohistochemistry staining.** TMA staining was performed using standard immunohistochemical staining procedures as previously described[50]. The HuR antibody (Santa Cruz) was used in a dilution of 1:200. To confirm the specificity of the primary antibody, tissue sections were incubated with control mouse IgG in the absence of primary antibody. The percentage and the intensity of positively stained cells were scored by inForm software and then verified by two pathologists independently in a blinded manner. The percentage of positive stained cells was scored as follows: 0, 1-25%, 26-75% and > 75%. The intensity of positive immunostaining was classified into four categories: 0, 1, 2, and 3 representing no visible staining; light brown, mid-brown and dark brown staining, respectively, with the same intensity covering more than 75% of the staining area. The intensity of cytoplasmic HuR staining is either 0 or 1. For the statistical analysis, the stained tumor tissues were divided into two groups: the low-expression group and the high-expression group. The low-expression group included tissues with negative staining or light staining with 1 + staining in ≤75% of cells; the high-expression group included tissues with intense staining with 1 + staining in > 75% of cells. Five specimens either damaged or not well stained were excluded, which left us 135 specimens for the analysis.

**Establishment of stable cell lines.** The GIPZ lentiviral short hairpin RNA (shRNA) constructs targeting human HuR (NM_001419.2) as well as GIPZ and TRIPZ empty vectors were purchased from Dharmacon. Same shRNA from GIPZ system were cloned into TRIPZ empty vector. The lentiCRISPRV2 vector was purchased from AddGene and the control single guide RNA (sgRNA) and HuR sgRNA were cloned into the vector as described[51,52]. The HuR lentiviral shRNA, sgRNA or control shRNA and sgRNA were co-transfected into HEK 293FT cells with the packaging plasmids pMD2.G and psPAX2 (both from AddGene). MDA-MB-231 and SUM159 cells were infected with virus medium and then selected with 1.5 μg/mL puromycin. Expression of inducible shRNA was induced by 0.5 μg/mL doxycycline. Single clones were generated by limiting dilution. Knock-down or knock-out of HuR expression was verified by qPCR and western blot.

**Invasion assay.** To analyze cell invasion, Corning BioCoat Matrigel Invasion Chambers (Bedford, MA) were used. (1–1.5) ×10⁵ cells in 0.5 mL of serum free medium were seeded in Matrigel-coated upper chambers and incubated for 22 h. Cells were fixed with 95% methanol and stained with 0.1% crystal violet. Non-invaded cells were removed from the upper surface of the membrane by cotton swabs. Cells that invaded were visualized and photographed with EVOS FL cell imaging systems (Life Technologies, Bothell, WA) under ×4 and ×20 magnification.

**Immunofluorescence.** Cells were seeded on 4-well Lab-TekII chamber slide (Fisher Scientific). Following fixation, permeabilization and blocking, cells were incubated with HuR antibody (Santa Cruz, 1:100 dilution) or mouse IgG followed with anti-mouse IgG-FITC antibody (Sigma). Nucleus were then stained with

DAPI. Images were taken with EVOS FL cell imaging systems (Life Technologies, Bothell, WA) under ×4 and ×20 magnification.

**Cellular thermal shift assay (CETSA).** CETSA were carried out as reported previously[53]. MDA-MB-231 cells were pre-treated with 20 μM KH-3 or DMSO for 1 h. Cells were then harvested and resuspended in PBS supplemented with complete protease inhibitor cocktail. The cell suspensions were aliquoted into PCR tubes and heated for 3 min at designated temperatures ranging from 42 to 54 °C followed by cooling at 25 °C for 3 min. Subsequently, cell suspensions were freeze-thawed three times using liquid nitrogen. The soluble fractions were then analyzed by western blot. The band intensities were measured using Image Studio Ver 4.0 (LI-COR Biosciences) and normalized to α-Tubulin.

**Fluorescence polarization assay.** The FP assay for HTS and dose–response curve was employed as previously described[17]. The only differences were that HTS was performed in 96-well black plates with a final volume of 100 μL and the compounds were screened at 20 μM concentration.

**RNA pull-down and RNP IP.** Two assays were carried out as previously described with minor modifications[17]. For RNA pull-down, cell lysate was incubated with 16-nt biotinylated Msi1 RNA oligo (1 μM), 18-nt biotinylated FOXQ1 oligo (5′-AU UAUUUAUAUAUUUUUG-3′, 1 μM) or random RNA oligo (1 μM) for 30 min with respective 10 μM unlabeled oligo, or DMSO, KH-3, KH-3B. For RNP IP, cells pre-treated with DMSO, KH-3B or KH-3 for 6 h were lysed and incubated with HuR antibody or mouse IgG (Sigma-Aldrich).

**Luciferase reporter assay.** For CDH1 reporter assay, HEK 293FT cells co-transfected with pGL3 vector with or without CDH1 promoter (Addgene, #61798) and renilla were treated with DMSO, KH-3 or KH-3B for 48 h. For FOXQ1 3′-UTR study, MDA-MB-231 and SUM159 cells transfected with pEZX-MT06 reporter vector with or without human FOXQ1 3′-UTR were treated with DMSO, KH-3 or KH-3B at indicated doses for 24 h. The cells were then harvested and assayed using the Dual-Glo Luciferase Assay (Promega). All firefly luciferase values were normalized to the renilla luciferase values. Relative luciferase activity was calculated for each treated sample by dividing normalized luciferase activity by that of DMSO control.

**Tumor invasion/metastasis qPCR array.** Total RNA isolated from MDA-MB-231 sgControl, HuR KO1, HuR KO2 and MDA-MB-231 cells treated with 10 μM KH-3 or DMSO were reverse transcribed into complementary DNA, and amplified by Human Tumor Invasion/Metastasis Primer Library (HTIM-I) from Real Time Primers LLC (Elkins Park), which contains 88 primer sets directed against invasion/metastasis related genes and 8 housekeeping gene primer sets as we described previously[54]. The heatmap was generated by Heatmapper (http://www.heatmapper.ca/) using Average Linkage as clustering method and Pearson as distance measurement method.

**RIP-seq and RNA-seq.** RNA immunoprecipited by HuR antibody or mouse IgG control from MDA-MB-231 cells and total RNA from MDA-MB-231 cells treated with DMSO or KH-3 were collected for library preparation. Each condition had two biological replicate. The libraries were sequenced using Illumina Hiseq 2500 system at Genome Sequencing Core at the University of Kansas. The RIP-seq dataset was mapped to the reference human genome hg38 using STAR. The corresponding peaks were then detected using HOMER[55] and annotated using ANNOVAR[56]. Finally, the RIP peaks that correspond to significant transcript abundance change were identified and compiled. The RNA-Seq dataset contains three groups: DMSO, 5 μM KH-3 and 10 μM KH-3. The DMSO group was considered as the control group. The reads were mapped against the transcript databases annotated in GENCODE v24[57] using STAR[58] default parameters. The read counts for each transcript were calculated using eXpress[59]. The differentially expressed transcripts were detected using DESeq2[60] with a q-value cutoff of 0.05.

**Animal study.** Female athymic NCr-nu/nu mice and BALB/c mice of 4–6 weeks old purchased from Charles River Laboratories were used for tumor formation and efficacy studies. $0.5 \times 10^6$ MDA-MB-231 cells or $1 \times 10^6$ SUM159 cells in 0.2 mL DMEM were inoculated to #2 mammary fat pad (MFP) of athymic nude mice. Tumor sizes were measured using a caliper twice a week. Tumor volume was calculated using the formula: (length × width²)/2, as we described previously[49]. Tumor growth inhibition (T/C %) is defined as the ratio of the mean tumor volume for the treated versus control group. For efficacy study in MFD model, mice were randomized into two groups when tumors reached ~50 mm³ and treated with 100 mg/kg KH-3 or vehicle control. The treatment was administrated via intraperitoneal injection three times per week for three weeks. For 2LMP experimental metastasis model, $0.5 \times 10^6$ 2LMP cells stably expressing luciferase in 0.3 mL DMEM were intravenously injected into tail veins of athymic nude mice. For 4T1 experimental metastasis model, $1 \times 10^4$ 4T1 cells stably expressing luciferase in 0.1 mL DMEM were intravenously injected into tail veins of BALB/c mice. Immediately following injection, mice were imaged by bioluminescence to assure wide

distribution and quality of the initial injection. The mice were then randomized into two groups and treated with 50 mg/kg KH-3 or vehicle control via intraperitoneal injection five times per week for five (2LMP model) or four (4T1 model) weeks. Bioluminescence imaging was taken weekly to monitor the metastasis burden at lung using our Carestream In-Vivo MS FS PRO Imaging System. When an image showed luminescent signaling at lung compared to background with automatic setting, the mouse was considered with tumor initiation. Imaging was taken twice a week thereafter first detectable signaling at lung. All animal experiments were carried out according to the protocol approved by the Institutional Animal Use and Care Committee at the University of Kansas.

**Statistics and reproducibility**. The GraphPad Prism 5.0 software was used for statistical analysis. The association between cytoplasmic HuR staining and clinicopathologic factors was assessed by using Fisher's exact test. The Kaplan–Meier method and the log-rank test were used to compare overall survival, defined as the time of patients from surgery until death, and distant disease-free survival, defined as the time of patients from surgery until having metastasis (patients alive were censored at the time of their last follow-up). Error bars in boxplots represent minimum and maximum values. Error bars in bar charts represent standard deviation (SD) except tumor growth data were expressed as mean ± standard error of mean (SEM). Student's $t$ test, one-way analysis of variance (ANOVA) followed by Tukey's post hoc test and two-way ANOVA followed by Bonferroni multiple comparison tests were employed to analyze the in vitro and in vivo data. The Kaplan–Meier method and the log-rank test were also used to analyze tumor initiation and overall mouse survival. A threshold $P < 0.05$ was defined as statistical significance. All in vitro experiments were repeated at least three time. Exact sample sizes and number of replicates were indicated in the figure legends.

**Reporting summary**. Further information on research design is available in the Nature Research Reporting Summary linked to this article.

## Data availability
Both the RIP-seq and RNA-seq data are deposited at Gene Expression Omnibus under accession number GSE129530. The source data underlying the graphs and charts presented in the main figures are shown as Supplementary Data 3. All other data supporting the findings of this study are available within the paper and Supplementary Information.

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

## Acknowledgements
The authors thank Jiajun Liu and Yu Zhan for their help on the animal studies and Jinan Wang for his suggestions on the computational docking. We thank Remya Ramesh for her help on synthesis of KH-3 and KH-3B. We thank Lisa Zhang for her help on H&E staining of lung tissues. We are grateful to the NCI/DTP Open Chemical Repository (http://dtp.cancer.gov) for providing chemical libraries.This study was supported in part by the National Institutes of Health grants R01 CA191785 (to L.X., J.A.) and R01 CA243445 (to L.X.), the Department of Defense Breast Cancer Research Program Level II grant BC151845 (to L.X., D.R.W.), the Susan G. Komen Career Catalyst Research grant CCR18548252 (to X.W.), the National Foundation for Cancer Research (to D.R.W.), Kansas Bioscience Authority Eminent Scholar Award (to D.R.W.), the National Cancer Institute Cancer Center Support Grant P30 CA168524 (to L.X. and D.R.W.) and Kansas Bioscience Authority Rising Star Award (to L.X.).

## Author contributions
X.W. contributed to experimental design, execution, and data analysis as well as the writing of this manuscript; G.G., L.L. and L.W. contributed to experimental execution, data analysis, and manuscript revision; S.H. contributed to scratch assay and invasion assay; C.Z. provided analysis and interpretation of RNA-Seq results; R.T.M. contributed to luciferase reporter assay; S.W. and S.R. synthesized compound KH-3 and KH-3B; R.G. performed computational docking; J.K. supervised computational docking and revised the manuscript; F.G. prepared HuR proteins; D.A.D. and D.R.W. assisted with experimental design and revised the manuscript; L.Li supervised tissue array study and revised the manuscript; M.J. provided material support and assisted with experimental design; J.A. supervised synthesis of KH-3 and KH-3B, assisted with experimental design and revised the manuscript; L.X. oversaw all experimental design, execution, analysis, interpretation, and communication of results as well as manuscript revision. All authors approved the final version of this manuscript.

## Competing interests
The authors declare no competing interests.
