## [Peer Review File · Communications Biology]

Reviewers' comments:

Reviewer #1 (Remarks to the Author):

In this manuscript, Wu et al. performed a screen to identify novel HuR inhibitor(s). By screening 2000 compounds, they identified KH-3 as a novel small molecule inhibitor of HuR. They demonstrated that this compound inhibits the ability of HuR to bind to a biotinylated AREMSi1 RNA oligomer. They additionally show that KH-3 prevents the binding of HuR to the previously identified mRNA targets BCL-2, XIAP and Musashi 2 in breast cancer cells. The decreased binding of HuR to these messages results in their decreased stability that, consequently, leads to a decrease in their protein levels. Importantly, they demonstrate that this novel inhibitor prevents the growth and progression of breast cancer cells both in vitro and in vivo. These results are similar to those observed when HuR is depleted in breast cancer cells suggesting that the effect of the inhibitor is due to its ability to target HuR. Lastly, the authors show that KH-3 mediates its effect on breast cancer growth/progression by preventing, in part, the binding of HuR to the FOXQ1 mRNA.

This is a very interesting manuscript that convincingly identifies KH-3 as a novel inhibitor of HuR. Indeed the authors demonstrate, in their manuscript, that KH-3 can prevent the growth and progression of breast cancer both in vitro and in vivo, by decreasing the binding of HuR to its target mRNAs such as FOXQ1. In general, the majority of the experiments provide sufficient evidence to support the conclusions set forth by the authors in the manuscript. Despite some concerns (listed below) the importance of their findings is underscored by the fact that they identify KH-3 as a novel therapeutic that could be used to prevent breast cancer growth and metastasis by targeting the function of HuR.

Comments:

1) In Figure 2h, the authors should provide quantifications with error bars and p-values in order to demonstrate that KH-3 significantly decreases the binding of HuR to the AREMSi1 RNA oligomer. Furthermore, they should demonstrate that the KH-3B inhibitor does not affect, in these pull down experiments, the binding of HuR to this oligomer. In addition, in Figure 2i, the authors should demonstrate that KH-3B does not prevent the association of HuR to its target mRNAs.

2) The stability experiments presented in Fig. 3c as well as the western blots shown in Fig. 3d should be presented with error bars and p values in order to demonstrate the significance of these results.

3) The authors demonstrate in Fig. 5a by performing RIP-coupled to RNA-Seq experiments that HuR associates with 2988 mRNAs in breast cancer cells. Interestingly, however, the authors demonstrated that although KH-3 modulates the expression of 3392 mRNAs, by total RNA Seq, in these cells, only 995 mRNAs of these were identified in the RIP-coupled to RNA-Seq experiments as HuR mRNA targets. The authors should reconcile why the KH-3 inhibitor did not affect the expression of all mRNAs identified in the RIP-coupled to RNA-Seq experiments.

4) The authors should show HuR protein levels in the western blot shown in Fig. 5b in order to assess if the expression of FOXQ1 in the various TNBC cell lines is correlated with HuR protein levels.

5) As mentioned above, the authors should provide quantifications with error bars and p-values for the RNA pull-down data presented in Fig. 5c.

6) As mentioned above, the western blots shown in Fig. 6e should be presented with error bars and p-values in order to demonstrate the significance of these results.

7) Several HuR inhibitors, including MS-444 and DHTS, have been shown to inhibit the function of HuR as well as the growth/progression of cancer cells both in vitro and in vivo (i.e Blanco et al., Oncotarget, 2016; D'Agostino, Sci Rep, 2015). A discussion of these studies, including a comparison/contrast to this present study should be provided in order for one to get a better understanding of the potential therapeutics which could be used to treated pathologies which occur due to the dysfunction of HuR.

Reviewer #2 (Remarks to the Author):

In this manuscript, Wu et al suggest that targeting the RNA-binding protein HuR may inhibit breast cancer invasion and metastasis. The authors provide a series of experiments and propose that HuR is upregulated in breast cancer. In particular, the authors suggest that high levels of its cytoplasmic form correlate with poor outcome and high-grade tumors. Through well-executed experiments, they provided a lead, KH3, to inhibit HuR function. The authors use RIP and RNA-sequencing to identify HuR and KH3 targets in breast cancer, identifying FOXQ1 as the potential mediator of HuR function targeted by KH3. Through the modulation of FOXQ1 translation and that of other proteins, KH3 may prevent HuR mediated contribution to breast cancer invasion. Overall, this is an interesting study providing incremental evidence in breast cancer over the well establish role of HuR and its contribution in other tumor types. This is a relevant contribution although some weaknesses have been identified. Whereas the information provided is relevant, the limitations must be address. Currently the observation is limited to a particularly experimental system in vivo, MDA-231 cells.

Conceptually, the manuscript builds on the idea of metastasis. Yet, besides some in vivo lung colonization assays in the last figure there is not a single evidence that the findings support this otherwise complex multi-step process. Invasion and migration are important cellular functions but they do not exclusively pertain to metastasis. Finally, lung colonization assays do not recap the metastatic process from the primary site. Thus, the language should be restricted to the evidence provided.

Overall is well established that the more cancer cells proliferate the more metastasis is detected, similarly the more tumor imitating properties the higher the likelihood to generate a tumor and a metastasis. In summary, central to the manuscript is whether HuR and its inhibition by KH3 target central tumor formation and expansion or truly metastasis functions. Figure 1 shows that HuR is necessary for proliferation, invasion and tumorigenesis. Please modify the text and conclusions accordingly.

Figure 1a and b are based on the IHC staining of of HuR and its location. Central to these analyses is to verify the antibody recognition of the desired protein. Please provide control and HuR knockout cell pellets IHC to confirm the staining specificity to the indicated protein. Given the intensity (Suppl figure 1) and narrow range of cytoplasmic staining a computerized scoring may provide more accuracy than human scoring.

Figure 1 evidence is pivotal to the manuscript, yet is all performed in one single cell line, MDA-231. An

effort to generalize the finding (as shown later) should be made to build on solid ground.

EMT and proliferation are negatively correlated. Interestingly, some of the observations herein seem to reinforce this phenomenon. To what extent is FOXQ1 uncoupling these two processes?

In figure 7 c, bioluminescence is used. However it is uncertain how metastasis scoring was performed, what was the threshold to call a lesion positive or negative, how was normalization of the lesions performed and how the lesions were compared to background. Otherwise, the definition of a metastasis may vary significantly from mouse to mouse and variations depend on the frequency of imaging follow up. In addition, representative bioluminescence images and H&E section of the lungs at the end points should be provided.

Reviewer #1:

Comments:

1) In Figure 2h, the authors should provide quantifications with error bars and p-values in order to demonstrate that KH-3 significantly decreases the binding of HuR to the ARE^{Msi1} RNA oligomer. Furthermore, they should demonstrate that the KH-3B inhibitor does not affect, in these pull down experiments, the binding of HuR to this oligomer. In addition, in Figure 2i, the authors should demonstrate that KH-3B does not prevent the association of HuR to its target mRNAs.

We appreciate the reviewer for this comment on the important technical details that we missed. We repeated the pull down experiments including KH-3B treatment. We also provided quantification data with error bars and p-values. As shown in the revised Fig. 2h, KH-3, but not KH-3B, significantly decreases the binding of HuR to the ARE^{Msi1} RNA oligomer. For the RNP IP, in fact, we did examine both KH-3 and KH-3B but only presented the results of KH-3 as KH-3B has no effect. Inspired by the reviewer's comment, we included the result of KH-3B in the revised Fig. 2i.

2) The stability experiments presented in Fig. 3c as well as the western blots shown in Fig. 3d should be presented with error bars and p values in order to demonstrate the significance of these results.

We agree with the reviewer and added p values in stability experiments as presented in Fig. 3c. We also did quantification of western blots shown in Fig. 3d (now as Fig. 3d and 3e) and presented with error bars and p values. We demonstrate that KH-3, but not KH-3B, significantly destabilizes Bcl-2, Msi2 and XIAP mRNAs and reduces the protein levels of these HuR targets in both MDA-MB-231 and SUM159 cell lines.

3) The authors demonstrate in Fig. 5a by performing RIP-coupled to RNA-Seq experiments that HuR associates with 2988 mRNAs in breast cancer cells. Interestingly, however, the authors demonstrated that although KH-3 modulates the expression of 3392 mRNAs, by total RNA Seq, in these cells, only 995 mRNAs of these were identified in the RIP-coupled to RNA-Seq experiments as HuR mRNA targets. The authors should reconcile why the KH-3 inhibitor did not affect the expression of all mRNAs identified in the RIP-coupled to RNA-Seq experiments.

We thank the reviewer for this insightful comment. When we analyzed the total RNA-seq data, the cut-off for differential expression was $q < 0.05$. The mRNAs modulated by KH-3 did not reach the cut-off ($q < 0.05$) were excluded. Additionally, as we show in Fig. 2h, 3c-e, the effect of KH-3 on different HuR targets such as Bcl-2, Msi2 and XIAP are not same. Therefore, as a small molecule inhibitor, KH-3 did not affect the expression of all mRNAs identified in RIP-seq experiments. We also state in the discussion: As a small molecule inhibitor of HuR, KH-3 phenocopies the HuR knockout in most of assays but less efficiently, which is possibly because KH-3 only modulates portion of HuR targets.

4) The authors should show HuR protein levels in the western blot shown in Fig. 5b in order to assess if the expression of FOXQ1 in the various TNBC cell lines is correlated with HuR protein levels.

We thank the reviewer for this insightful comment. In fact, we did examine HuR protein level as well in the various TNBC cell lines, but only presented FOXQ1 and E-cadherin protein levels as

we focused on the correlation between FOXQ1 and E-cadherin. Inspired by the reviewer's comment, now we include HuR protein level in the revised Fig. 5b. There is no obvious correlation between HuR and FOXQ1 protein expression as HuR is expressed in all TNBC lines tested. We also tried to examine cytoplasmic HuR expression level since regulation of RNA stability and translation is mainly related to cytoplasmic localization of HuR. However, cytoplasmic HuR expression was weak or undetectable in most TNBC cell lines under normal (unstressed) condition.

5) As mentioned above, the authors should provide quantifications with error bars and p-values for the RNA pull-down data presented in Fig. 5c.

We agree with the reviewer. In the revised Fig. 5c (right panel), quantifications with error bars and p-values are presented. KH-3, but not KH-3B, significantly decreased the binding of HuR to the ARE^{FOXQ1} RNA oligomer.

6) As mentioned above, the western blots shown in Fig. 6e should be presented with error bars and p-values in order to demonstrate the significance of these results.

We appreciate the reviewer for this comment. In the revised Fig. 6e (bottom), quantifications with error bars and p-values are presented. The inhibitory effect of KH-3 on other HuR target proteins was not affected by FOXQ1 overexpression. Exogenous FOXQ1 abolished the inhibitory effect of KH-3 on FOXQ1 protein levels.

7) Several HuR inhibitors, including MS-444 and DHTS, have been shown to inhibit the function of HuR as well as the growth/progression of cancer cells both in vitro and in vivo (i.e Blanco et al., *Oncotarget*, 2016; D'Agostino, *Sci Rep*, 2015). A discussion of these studies, including a comparison/contrast to this present study should be provided in order for one to get a better understanding of the potential therapeutics which could be used to treated pathologies which occur due to the dysfunction of HuR.

We agree with the reviewer that our discussion on the other HuR inhibitors was insufficient. We added more discussion and comparison on other two published HuR inhibitors (MS-444 and DHTS), which were tested in HCT116 xenograft model in vivo.

Reviewer #2:

Whereas the information provided is relevant, the limitations must be address. Currently the observation is limited to a particularly experimental system in vivo, MDA-231 cells.

We thank the reviewer for this insightful and inspiring comment. We included another cell line SUM159 with HuR knockout by CRISPR. Knockout HuR in SUM159 cells showed similar results to those in MDA-MB-231 cells in most of experiments (supplementary Fig. 4). We also included a 4T1 experimental metastasis model. KH-3 treatment improved the survival time of mice in this model (supplementary Fig. 8d).

Conceptually, the manuscript builds on the idea of metastasis. Yet, besides some in vivo lung colonization assays in the last figure there is not a single evidence that the findings support this otherwise complex multi-step process. Invasion and migration are important cellular functions but they do not exclusively pertain to metastasis. Finally, lung colonization assays do not recap the metastatic process from the primary site. Thus, the language should be restricted to the evidence provided.

We agree with the reviewer that lung colonization assays do not recap the metastatic process from the primary site. We revised our text to more precisely describe the findings based on the results provided.

Overall it is well established that the more cancer cells proliferate the more metastasis is detected, similarly the more tumor imitating properties the higher the likelihood to generate a tumor and a metastasis. In summary, central to the manuscript is whether HuR and its inhibition by KH3 target central tumor formation and expansion or truly metastasis functions. Figure 1 shows that HuR is necessary for proliferation, invasion and tumorigenesis. Please modify the text and conclusions accordingly.

We thank the reviewer for this insightful comment. As HuR is involved in multiple cancer hallmarks, inhibition of HuR by KH-3 decreases proliferation, invasion and tumorigenesis. However, this manuscript mainly focuses on the HuR function on invasion and metastasis. We revised the text and conclusions accordingly.

Figure 1a and b are based on the IHC staining of HuR and its location. Central to these analyses is to verify the antibody recognition of the desired protein. Please provide control and HuR knockout cell pellets IHC to confirm the staining specificity to the indicated protein. Given the intensity (Suppl figure 1) and narrow range of cytoplasmic staining a computerized scoring may provide more accuracy than human scoring.

We thank the reviewer for this insightful and inspiring comment. We did immunofluorescence staining in both control and HuR KO cells using the same HuR antibody used for IHC. Only control cells incubated with HuR antibody showed positive staining of HuR predominantly localized in the nuclei (supplementary Fig. 3). For the scoring of IHC staining, in fact, the staining was scored by computer program (inForm) first and then verified by two pathologists independently. We did not state clearly in the method and have revised accordingly.

Figure 1 evidence is pivotal to the manuscript, yet is all performed in one single cell line, MDA-231. An effort to generalize the finding (as shown later) should be made to build on solid ground.

We thank the reviewer for this insightful and inspiring comment. We included another TNBC cell line SUM159 in our revised manuscript. Knockout HuR in SUM159 cells showed similar results to those in MDA-MB-231 cells in most of experiments (supplementary Fig. 4).

EMT and proliferation are negatively correlated. Interestingly, some of the observations herein seem to reinforce this phenomenon. To what extent is FOXQ1 uncoupling these two processes?

We apologize for making reviewer confused. HuR involves in both EMT and proliferation hallmarks. However, for FOXQ1, we only showed that FOXQ1 overexpression abolished upregulated CDH1 induced by HuR KO or KH-3 treatment, and FOXQ1 overexpression rescued the phenotype of cell invasion impaired by HuR KO or KH-3 treatment. We stated that FOXQ1 is one of HuR targets that contributes to HuR's role in breast cancer invasion and metastasis, potentially through promoting EMT. Proliferation inhibition by HuR KO or KH-3 may work through other HuR targets.

In figure 7 c, bioluminescence is used. However it is uncertain how metastasis scoring was performed, what was the threshold to call a lesion positive or negative, how was normalization of the lesions performed and how the lesions were compared to background. Otherwise, the

definition of a metastasis may vary significantly from mouse to mouse and variations depend on the frequency of imaging follow up. In addition, representative bioluminescence images and H&E section of the lungs at the end points should be provided.

We thank the reviewer for this insightful and inspiring comment. We agree that the description of how to perform bioluminescent imaging and how to determine the initiation of lung colonization was not clear. We have revised the text accordingly. Imaging was taken once a week at the beginning. Once the first mouse had detectable signaling in lung, imaging was taken twice a week thereafter. When an image showed signal in lung compared to background with automatic setting of our Carestream In-Vivo MS FS PRO Imaging System, the mouse was considered with tumor initiation. Representative bioluminescence images were provided to present initiation and progression of lung metastasis. Representative images of H&E staining of the lungs were also provided for validation.

Reviewer #3

Overall comments

Therapeutic targeting of RNA-binding proteins (RBPs) is a relatively new strategy in cancer treatment and HuR is a sensible target as it is one of the most frequently dysregulated RBPs in tumors. Indeed, the authors convincingly demonstrate that KH-3 has therapeutic value in inhibiting primary tumor growth. However the effect of KH-3 treatment on lung metastasis could be explained by reduced primary tumor size, and its mechanism of action is unclear. The authors argue that KH-3 functions at least partly through inhibition of the HuR-FOXQ1 mRNA interaction, but the data suggest that KH-3 reduces FOXQ1 expression indirectly. KH-3 treatment only modestly reduces the interaction between HuR and AREFOXQ1 (Fig 5C) and does not affect the stability of FOXQ1 mRNA (Fig S4), which is inconsistent with a model in which the compound directly interferes with the RBP-mRNA interaction. Furthermore, it takes hours for KH-3 treatment to reduce the mRNA levels of FOXQ1 (4 hrs for SUM159 and 16 hrs for MDA-MB-231), suggesting that any effect KH-3 has on FOXQ1 expression is an indirect one downstream of HuR inhibition or possibly off-target effects. Considering the IC₅₀ for KH-3 in HuR KO cells is only 2-fold higher compared to wildtype cells (Fig 3B), off-target effects seem highly likely. In the conclusion section, the authors claim to “identify a novel inhibitor KH-3 that disrupts HuR-mRNA interactions through competitively binding to HuR and therefore blocks HuR functions, leading to the decay of HuR target mRNAs and reduction of translation.” I would argue that this is an over-interpretation of the data, considering that there is little evidence that the HuR-FOXQ1 interaction is disrupted by KH-3 and the authors did not provide any data directly relating to the translation of HuR targets (protein levels are affected by much more than just translation rate so a Western blot is not really sufficient to justify this claim). Also in the conclusion section, the authors present a model for how KH-3 affects the HuR pathway through FOXQ1 and other proteins including LAMA4 and beta-catenin. I believe there is only sufficient data in the present study to support the FOXQ1-CDH1 arm of this model. Overall, this study clearly establishes a rationale for targeting HuR and demonstrates potential therapeutic value in KH-3, but does not persuasively link the two ideas mechanistically.

We thank the reviewer for these critical, insightful and inspiring comments. The model we used to evaluate anti-metastatic potential of KH-3 is an experimental metastasis model without primary tumor. So the effect of KH-3 treatment on lung metastasis could be explained by reduced/delayed lung colonization. We agree that KH-3 does not affect the stability of FOXQ1 but KH-3 does reduce the mRNA levels (Fig. 5e) and protein levels (Fig. 6e) of FOXQ1. To investigate whether the inhibitory effect on FOXQ1 is dependent on HuR or not, we performed additional experiments using HuR KO cells. As shown in the supplementary Fig. 7b, KH-3 reduces more

than 60% and 80% of FOXQ1 mRNA at doses of 5 μ M and 10 μ M respectively in control cells while KH-3 decreases less than 20% and 40% of FOXQ1 mRNA at dose of 5 μ M and 10 μ M respectively in HuR KO cells. HuR KO significantly attenuated the inhibitory effect of KH-3 on FOXQ1 mRNA, indicating that this inhibitory effect of KH-3 on FOXQ1 mRNA was at least partially dependent on HuR. Moreover, as listed in several HuR review papers (e.g., *Wiley Interdiscip Rev RNA* 1, 214 (2010), *Neoplasia* 18, 674 (2016)), the effect of HuR on mRNA is mRNA stabilization and/ or translation modulation. For some mRNAs, HuR affects both mRNA stability and translation. For some mRNAs, HuR affects either mRNA stability or translation. In our case, although KH-3 does not affect the stability of FOXQ1 mRNA, it affects the stability of Bcl-2, Msi2 and XIAP mRNAs. We agree with the reviewer that more evidence are needed to include LAMA4 and β -catenin in our working model of KH-3. We have revised the text and working model accordingly.

Specific comments

1. Fig S1A: Higher quality images of the IHC would be helpful (I cannot tell the difference between the 3 pictures in terms of cytoplasmic staining)

We apology for that. Due to the high intensity of nucleic staining and low intensity cytoplasmic staining, the difference among no, low and high cytoplasmic staining is not very apparent. We replaced with images with higher resolution.

2. Fig 1G-H: The in vivo experiment was only performed with 1 of 2 HuR KO cell lines. However, the cell line used does not form primary tumors and therefore cannot be evaluated in terms of metastasis. The second KO cell line or the dox-inducible shRNA cell line should be tested as well. This is an important point because the whole rationale for targeting HuR is that it promotes metastasis and the only data to support this are from in vitro experiments.

We thank the reviewer for this insightful and inspiring comment. We included another cell line SUM159 with HuR knockout. SUM159 cells with HuR knockout form tumors in vivo but it takes much longer time, and the tumor sizes are significantly smaller than those formed from sgControl cells.

3. Fig 3D: I am curious as to how similar KH-3 treatment is to HuR KO in the context of HuR's targets. Also a quantification of this blot and others would be helpful for interpretation.

We thank the reviewer for this inspiring comment. We included the protein levels of Bcl-2, Msi2 and XIAP in control and HuR KO clones (Fig. 2c and supplementary Fig. 4a). We also provided quantification of western blots with error bar and p values. KH-3 reduced the protein levels of HuR targets similarly to HuR KO.

4. Fig 4C: This panel shows qPCR data from a subset of invasion/metastasis related genes, some of which are metastasis promoters and some of which are suppressors. It would be helpful to know which genes correspond to which category (perhaps this could be achieved through patterning of the bars) and why this subset was chosen out of 88 genes. Again I am interested in how this data compares to HuR KO cells. A heatmap comparing expression of these genes in control vs KH-3 vs HuR KO could be useful.

We thank the reviewer for this insightful and inspiring comment. Inspired by the review, we presented the qPCR data as a heatmap including all 79 detectable genes in control, KH-3 and HuR KO. Clustering among different samples indicates that the gene profile for KH-3 treated samples is substantially similar to that for HuR KO1.

5. Fig S4 and Fig 5E-F: KH-3 does not affect the stability of FOXQ1 mRNA and only affects its levels after at least 4 hours. Could it be instead affecting FOXQ1 mRNA localization or translation?

We thank the reviewer for this insightful and inspiring comment, which is definitively worth further study. It would be interesting to investigate how KH-3 inhibits FOXQ1 mRNA at the sub-cellular level, since it does not affect the stability.

6. The authors suggest in their model that KH-3 functions by inhibiting EMT, however only 2 EMT-related genes were investigated. Is an EMT signature negatively correlated with KH-3 mRNAseq data? This would bolster the argument.

We thank the reviewer for this insightful and inspiring comment. Inspired by the review, we did Gene set enrichment analysis (GSEA) for RNA-seq data focusing on epithelial-mesenchymal transition hallmark set. The analysis displays that EMT genes were negatively enriched in KH-3 treated samples (Supplementary Fig. 6c).

7. I was surprised to find new pieces of data in the conclusion section. Is there some place within the results section where they might fit?

We apology for that. These data were moved to results section.

REVIEWERS' COMMENTS:

Reviewer #1 (Remarks to the Author):

The authors have addressed all of my concerns/comments in their revised manuscript.

Reviewer #2 (Remarks to the Author):

Overall, the authors have addressed most of my comments providing significant text changes where needed as well as new data. In particular, experiments with 4T1 and SUM159 provide generalisation of the findings and robustness. In addition, clarification of how metastasis was scored on the basis of luciferase activity was central as per reproducibility purposes. Finally, the improved statistical analyses, and clarification of the mechanisms provide further granularity. There may be some interpretation subtleties that this reviewer may not agree with but this is fair and should not preclude from publication.

The manuscript has significantly improved through the revision process.

Reviewer #3 (Remarks to the Author):

The authors have addressed my concerns. I would suggest performing hierarchical clustering on the new heat map so the similarities between KH-3 treatment and HuR KO are more apparent.